# Jak-Stat pathway induces *Drosophila* follicle elongation by a gradient of apical contractility

**Hervé Alégot, Pierre Pouchin, Olivier Bardot, Vincent Mirouse***

GReD Laboratory, Université Clermont Auvergne - CNRS UMR 6293- INSERM U1103, Clermont-Ferrand, France

**Abstract** Tissue elongation and its control by spatiotemporal signals is a major developmental question. Currently, it is thought that *Drosophila* ovarian follicular epithelium elongation requires the planar polarization of the basal domain cytoskeleton and of the extra-cellular matrix, associated with a dynamic process of rotation around the anteroposterior axis. Here we show, by careful kinetic analysis of *fat2* mutants, that neither basal planar polarization nor rotation is required during a first phase of follicle elongation. Conversely, a JAK-STAT signaling gradient from each follicle pole orients early elongation. JAK-STAT controls apical pulsatile contractions, and Myosin II activity inhibition affects both pulses and early elongation. Early elongation is associated with apical constriction at the poles and with oriented cell rearrangements, but without any visible planar cell polarization of the apical domain. Thus, a morphogen gradient can trigger tissue elongation through a control of cell pulsing and without a planar cell polarity requirement.
DOI: https://doi.org/10.7554/eLife.32943.001

## Introduction

Tissue elongation is an essential morphogenetic process that occurs during the development of almost any organ. Therefore, uncovering the underlying molecular, cellular and tissue mechanisms is an important challenge. Schematically, tissue elongation relies on at least three determinants. First, the elongation axis must be defined by a directional cue that usually leads to the planar cell polarization (pcp) of the elongating tissue. Second, a force producing machinery must drive the elongation, and this force can be generated intrinsically by the cells within the elongating tissue and/or extrinsically by the surrounding tissues. Finally, such force induces tissue elongation via different cellular behaviors, such as cell intercalation, cell shape modification, cell migration or oriented cell division. This is exemplified by germband extension in *Drosophila* embryos where Toll receptors induce Myosin II planar polarization, which drives cell rearrangements (*Bertet et al., 2004*; *Irvine and Wieschaus, 1994*; *Blankenship et al., 2006*; *Paré et al., 2014*).

In recent years, *Drosophila* egg chamber development has emerged as a powerful model to study tissue elongation (*Bilder and Haigo, 2012*; *Cetera and Horne-Badovinac, 2015*). Each egg chamber (or follicle) consists of a germline cyst that includes the oocyte, surrounded by the follicular epithelium (FE), a monolayer of somatic cells. The FE apical domain faces the germ cells, while the basal domain is in contact with the basement membrane. Initially, a follicle is a small sphere that progressively elongates along the anterior-posterior (AP) axis, which becomes 2.5 times longer than the mediolateral axis (aspect ratio [AR] = 2.5), prefiguring the shape of the fly embryo.

All the available data indicate that follicle elongation relies on the FE. Specifically, along the FE basal domain, F-actin filaments and microtubules become oriented perpendicularly to the follicle AP axis (*Gutzeit, 1990*; *Viktorinová and Dahmann, 2013*). The cytoskeleton planar polarization depends on the atypical cadherin Fat2, which acts via an unknown mechanism (*Viktorinová et al.,*

***For correspondence:**
vincent.mirouse@u-clermont1.fr

**Competing interests:** The authors declare that no competing interests exist.

2009; *Viktorinová and Dahmann, 2013*; *Chen et al., 2016*). Fat2 is also required for a dynamic process of collective cell migration of all the follicle cells around the AP axis until stage 8 of follicle development. This rotation reinforces F-actin planar polarization and triggers the polarized deposition of extracellular matrix (ECM) fibrils perpendicular to the AP axis (*Haigo and Bilder, 2011*; *Lerner et al., 2013*; *Viktorinová and Dahmann, 2013*; *Cetera et al., 2014*; *Isabella and Horne-Badovinac, 2016*; *Aurich and Dahmann, 2016*). These fibrils have been proposed to act as a molecular corset, mechanically constraining follicle growth along the AP axis during follicle development (*Haigo and Bilder, 2011*). In addition, Fat2 is required for the establishment of a gradient of basement membrane (BM) stiffness at both poles at stage 7–8 (*Crest et al., 2017*). This gradient also depends on the morphogen-like activity of the JAK-STAT pathway, and softer BM near the poles would allow anisotropic tissue expansion along the A-P axis (*Crest et al., 2017*). After the end of follicle rotation, F-actin remains polarized in the AP plane during stages 9–11 and follicular cells (FCs) undergo oriented basal oscillations that are generated by the contractile activity of stress fibers attached to the basement membrane ECM via integrins (*Bateman et al., 2001*; *Delon and Brown, 2009*; *He et al., 2010*).

Nonetheless, in agreement with recently published observations, we noticed that a first phase of follicle elongation does not require *fat2* and the planar polarization of the basal domain (*Aurich and Dahmann, 2016*). We therefore focused on this phase, addressing main three questions which are: how the follicle elongation axis is defined, what the molecular motor triggering elongation in a specific axis is, and how FCs behave during this phase.

## Results

### Polar cells define the axis of early elongation

We analyzed the follicle elongation kinetics in $fat2^{58D}$ mutants, which block rotation and show a strong round-egg phenotype. Follicle elongation is normal in *fat2* mutants during the first stages (3–7) with an AR of 1.6 (*Figure 1a–d*). Thus, at least two mechanistically distinct elongation phases control follicle elongation, a first phase (stages 3–7), which is independent of *fat2*, rotation and ECM basal polarization, and a later phase (stages 8–14), which requires *fat2*. This observation is consistent with the absence of an elongation defect of clonal loss-of-function of *vkg* before stages 7–8 (*Bilder and Haigo, 2012*).

To try to identify the mechanism that regulates the early phase of follicle elongation, we first analyzed trans-heterozygous *Pak* mutant follicles, which never elongate (*Conder et al., 2007*) (*Figure 1e*). The *Pak* gene encodes a Pak family serine/threonine kinase that localizes at the FE basal domain. *Pak* mutants also show many other abnormalities, such as the presence of more than one germline cyst and abnormal interfollicular filaments ([*Vlachos et al., 2015*] and not shown). Interfollicular cells derive from prepolar cells that also give rise to the polar cells, which prompted us to analyze the distribution of the latter using the specific marker FasIII (*Bastock and St Johnston, 2008*; *Horne-Badovinac and Bilder, 2005*). Polar cells are pairs of cells that differentiate very early and are initially required for germline cyst encapsulation (*Grammont and Irvine, 2001*). They also have a role as an organizing center for the differentiation of FC sub-populations during mid-oogenesis (*Xi et al., 2003*). In WT follicles, polar cells are localized at the follicle AP axis extremities (*Figure 1b*). Conversely, in *Pak* mutants, we observed a single polar cell cluster or two clusters close to each other (*Figure 1e*). This suggests that *Pak* is required for polar cell positioning, although a role in the specification or survival of these cells cannot be excluded, which in turn could play a role in defining the elongation axis. Some dominant suppressors of the *Pak* elongation defect have been identified, including PDGF- and VEGF-receptor related (Pvr), although the reason for this suppression is unknown (*Vlachos and Harden, 2011*). By using flies that are heterozygous for a *Pvr* allele and mutant for *Pak*, we observed that the normal positioning of polar cells is frequently but not always restored (*Figure 1f* and *Figure 1—figure supplement 1c*). We quantitatively compared the elongation of those two situations, normal or abnormal polar cells, by plotting the long axis as a function of the short axis for previtellogenic stages (before stage 8) and determined the corresponding regression line (*Figure 1—figure supplement 1d*). We defined an elongation coefficient that corresponds to the slope of this line and for which a value of 1 means no elongation. This method allows us to quantify elongation independently of any bias that could be introduced by stage

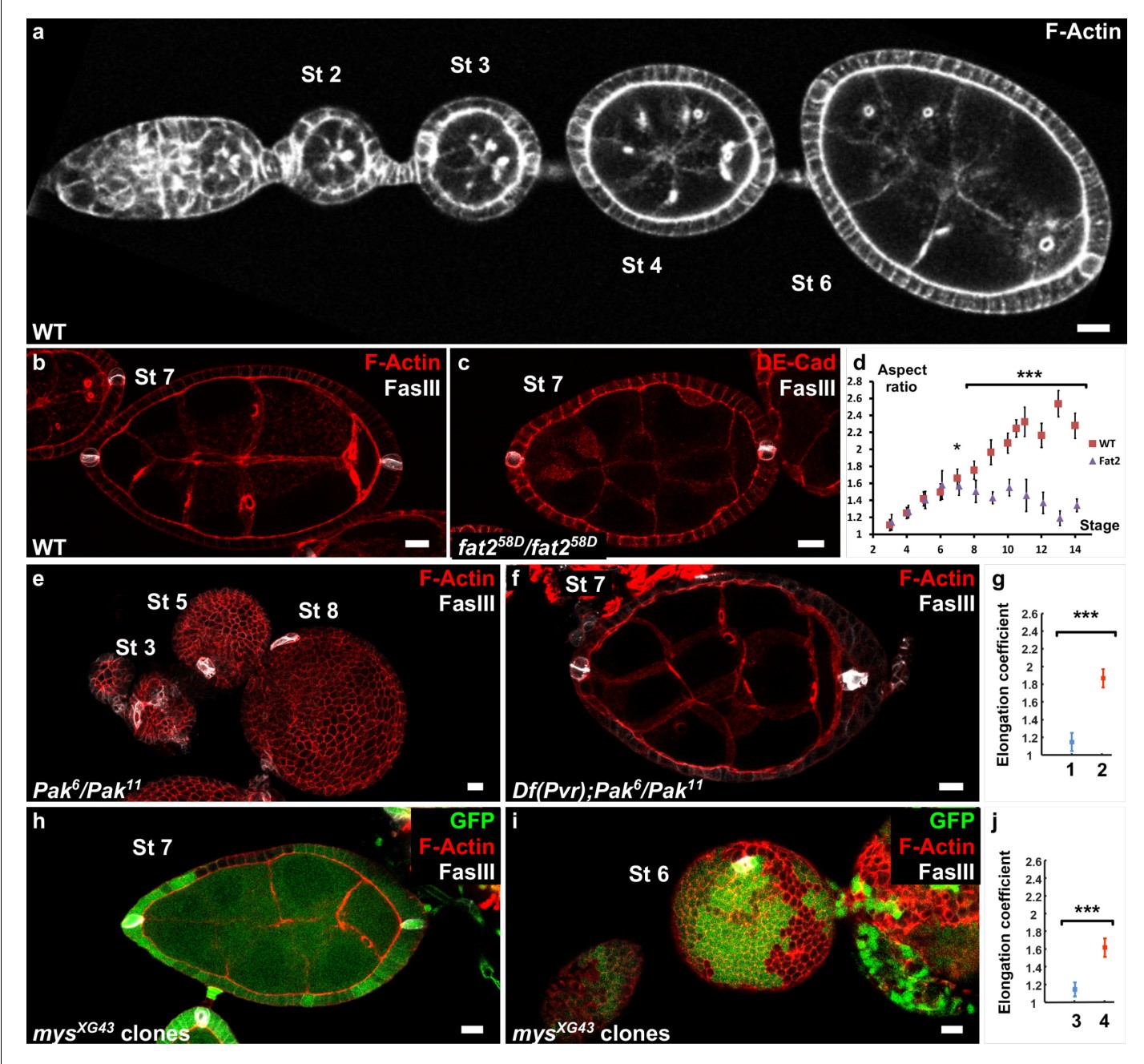

**Figure 1.** Polar cells determine the axis of early elongation.  (a) WT ovariole illustrating follicle elongation during the early stages of oogenesis (stages 2–6). (b) Optical cross-section of a stage 7 WT follicle stained with FasIII, a polar cell marker (white) and F-actin (red). (c) Stage seven *fat2* mutant follicle stained with FasIII (white) and DE-Cad (red). (d) Elongation kinetics of WT and *fat2* mutant follicles (n > 6 for each point). (e) Z-projection of a *Pak* mutant ovariole. Round follicles have only one cluster of polar cells (stage 5 and 8 follicles) or two non-diametrically opposed clusters (stage 3 follicle). (f) Removing a copy of *Pvr* restores early elongation and polar cell position in *Pak* mutants. (g) Elongation coefficient of *Pak⁶/Pak¹¹*, *Df(Pvr)/+* follicles, affecting (n = 23) (1) or not (n = 13) (2) polar cell positioning. (h,i) View of a *mys* mutant clone (GFP-negative) in a mosaic follicle showing (h) normal polar cell positioning and no elongation defect and (i) abnormal polar cell positioning and an early elongation defect. (j) Elongation coefficient of follicles containing mutant clones for *mys* affecting (n = 34) (3) polar cell positioning or not (n = 31) (4). Full details of the genotypes and sample sizes are given in the supplementary files. (p *<0.05, **<0.01, ***<0.001.) Scale bars are 10 μm throughout.

DOI: https://doi.org/10.7554/eLife.32943.002

The following figure supplement is available for figure 1:

**Figure supplement 1.** Polar cells determine the axis of early elongation.

DOI: https://doi.org/10.7554/eLife.32943.003

determination approximation due to aberrant follicle shape or differentiation. Moreover, focusing on previtellogenic stages allows the exclusion of genotypes that affect only the late elongation phase. It is exemplified by a *fat2* mutant that does not induce significant defects if we include only stage 3–7 follicles (previtellogenic), but does show a difference if we include stage 8 follicles (*Figure 1—figure supplement 1a,b*). The statistical comparison of the elongation coefficients clearly shows that restoring polar cell position by removing one copy of *Pvr* in *Pak* mutants strongly rescues follicle elongation (*Figure 1g* and *Figure 1—figure supplement 1c,d*).

Although not been fully demonstrated in this context, Pak often works as part of the integrin signaling network, and mosaic follicles containing FC clones that are mutant for *myospheroid* (*mys*), which encodes the main fly β-integrin subunit, also show a round follicle phenotype at early stages (*Haigo and Bilder, 2011*). We noticed that in some follicles containing *mys* mutant clones, polar cells are mispositioned, a defect generally observed when at least one polar cell is mutant. As in *Pak* mutants, the two polar cell clusters are not diametrically opposed (*Figure 1—figure supplement 1e*), or a single cluster is observed (*Figure 1i*, *Video 1*). Importantly, the polar cell positioning defect is associated with the round follicle phenotype (*Figure 1j*, and *Figure 1—figure supplement 1f*). Conversely, in mosaic follicles in which polar cell positioning was not affected, the round egg phenotype is never observed at early stages, even with large mutant clones (*Figure 1h* and *Figure 1—figure supplement 1f*, *Video 2*). In agreement, the elongation coefficient of mosaic follicles that have normal polar cell positioning is much higher than that for those with abnormal polar cells (*Figure 1j*). Thus, together, these results indicate that *pak* and *mys* mutants are not required for the early phase of elongation once polar cells are well-placed and thus affect this phase indirectly. The results also strongly suggest that polar cells are required to define the follicle elongation axis.

## A JAK-STAT gradient from the poles is the cue for early elongation

Once the follicle is formed, polar cells are important for the differentiation of the surrounding FCs. From stage 9 of oogenesis, FCs change their morphology upon activation by Unpaired (Upd), a ligand for the JAK-STAT pathway, which is exclusively produced by polar cells throughout oogenesis (*Silver and Montell, 2001*; *Xi et al., 2003*; *McGregor et al., 2002*). To identify the FCs in which the JAK-STAT pathway is active, we used a reporter construct in which GFP transgene expression is controlled by STAT binding repeat elements in the promoter (*Bach et al., 2007*). During the early stages of oogenesis, the pathway is active in all the main body FCs (*Figure 2b*). Moreover, we observed differences in GFP expression level (and thus STAT activity) between the poles and the mediolateral region, starting at about stage 3, concomitantly with the beginning of elongation (*Figure 2b,h*). At later stages (5–7), these expression differences lead to the formation of a gradient of STAT activity, as indicated by strong GFP expression at each pole and very weak or no signal in the large mediolateral part of each follicle (*Figure 2b,h*, and *Figure 2—figure supplement 1a*). Thus, the spatiotemporal pattern of JAK-STAT activation is consistent with a potential role of this pathway in follicle elongation.

The key role of JAK-STAT signaling during follicle formation precluded the analysis of elongation defects in large null mutant clones

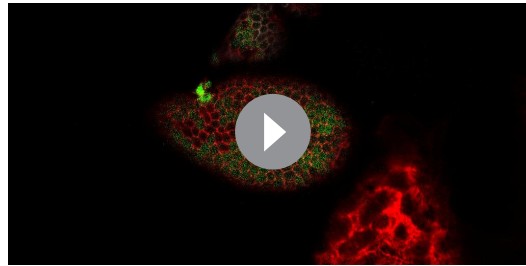

**Video 1.** Full z-stack of a follicle with a *mys* mutant clone that affects polar cells.
DOI: https://doi.org/10.7554/eLife.32943.004

**Video 2.** Full z-stack of a follicle with a *mys* mutant clone that does not affect polar cells.
DOI: https://doi.org/10.7554/eLife.32943.005

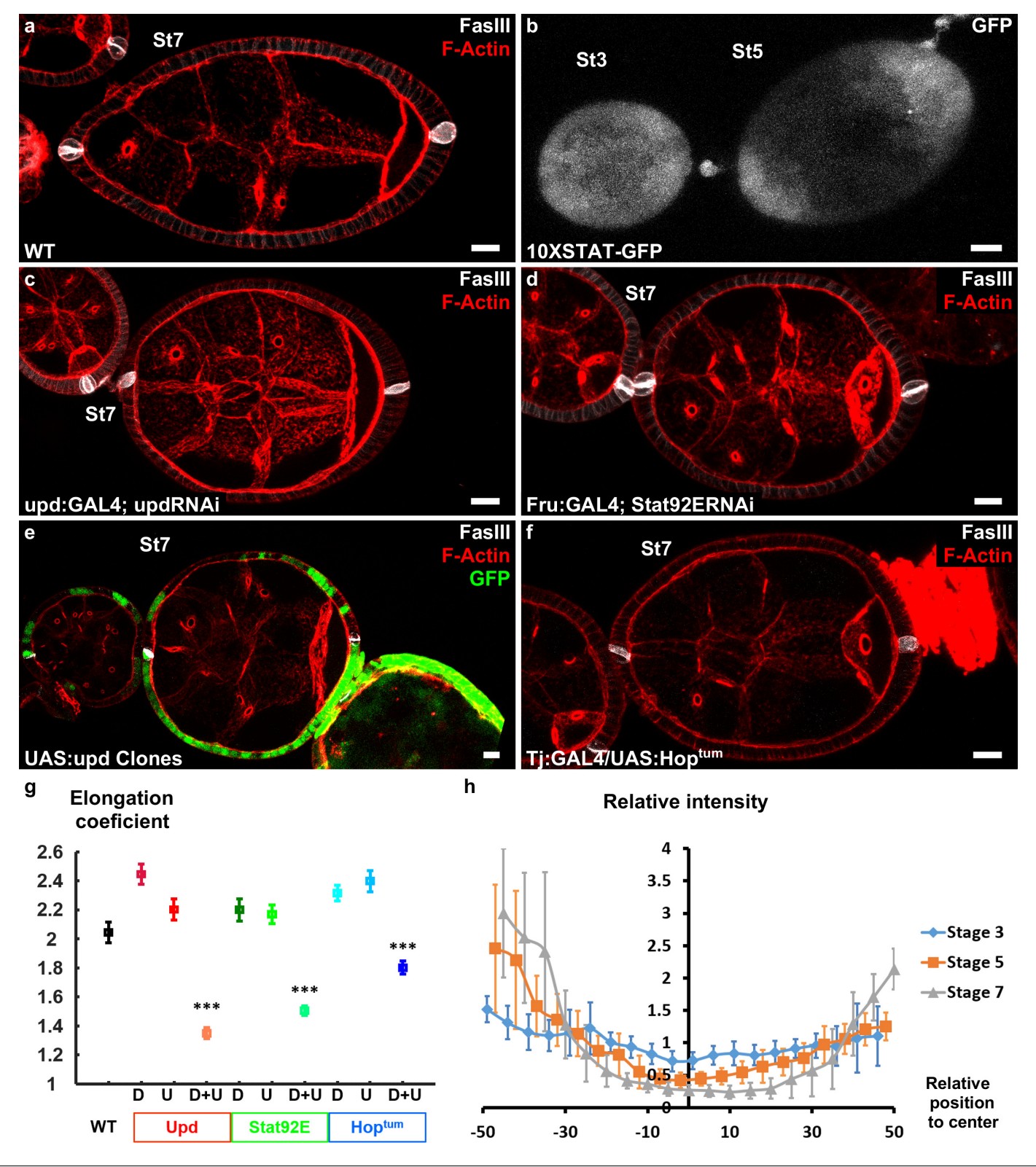

**Figure 2.** Upd is a polarizing cue for early elongation. (a) Optical cross-section of a WT stage 7 follicle stained with FasIII (polar cell marker, white) and F-actin (red). (b) Expression of the 10xStatGFP reporter showing the progressive formation of a STAT gradient at each pole. Early elongation is affected by (c) knocking down *upd* in polar cells or (d) the expression of *Stat92E* in the anterior and posterior follicular cells. Early elongation is also affected by (e) clonal ectopic expression of upd (GFP-positive cells) and by (f) expression of a *Hop* gain of function mutant in all follicular cells. (g) Quantification of
*Figure 2 continued on next page*

*Figure 2 continued*

the elongation coefficient in WT and the different JAK-STAT loss- and gain-of-function genotypes — corresponding to Upd:GAL4 and Upd-RNAi, Fru: GAL4 and STAT92E-RNAi, and Tj:GAL4 and UAS:Hop^tum — during early and intermediate stages of elongation (D, Driver; U, UAS line). (For each point n > 30; p *<0.05, **<0.01, ***<0.001.) (**h**) Quantification of the Stat activity gradient at stages 3, 5 and 7 using the 10XSTATGFP reporter. A gradient is already visible at stage 3 and becomes more visible until stage 7. Scale bars are 10 µm throughout. Relative intensity = intensity at a given position/ mean intensity of measured signal. Full details of the genotypes and sample sizes are given in the supplementary files.

DOI: https://doi.org/10.7554/eLife.32943.006

The following figure supplement is available for figure 2:

**Figure supplement 1.** JAK-STAT singalling and early elongation.

DOI: https://doi.org/10.7554/eLife.32943.007

(*McGregor et al., 2002*). Therefore, we knocked-down by RNAi the ligand *upd* and the most downstream element of the cascade, the transcription factor *Stat92E*, both efficiently decreasing the activity of the pathway in the follicular epithelium (*Figure 2a,c,d,g* and *Figure 2—figure supplement 1*). *Upd* knockdown was performed using either upd:GAL4 that is specifically expressed in the polar cells (*Khammari et al., 2011*) or tj:GAL4 that is expressed in all FCs, and then analyzed only in follicles that contained one germline cyst and correctly placed polar cells. During the early stages, with both drivers, such follicles are significantly rounder than control follicles (*Figure 2a,c,g* and *Figure 2—figure supplement 1e*). This indicates a role for JAK-STAT pathway in early elongation and confirms the causal link between polar cells and early elongation. Moreover, knockdown of *Stat92E* using a driver that is specifically expressed at the poles (Fru:GAL4) also affects early elongation (*Figure 2d,g* and *Figure 2—figure supplement 1e*), suggesting a transcriptional control of elongation by JAK-STAT (*Borensztejn et al., 2013*). These results are the first examples of loss of function with an effect only on early elongation and independent of polar cells position, and indicate that both Upd secreted by the polar cells and JAK-STAT activation in FCs arerequired for follicle elongation. Moreover, clonal ectopic *upd* overexpression completely blocks follicle elongation, without affecting polar cell positioning (*Figure 2e*), demonstrating that Upd is not only a prerequisite for the elongation but the signal that defines its axis (n = 20). Similarly, general expression of Hop^Tum, a gain-of-function mutation of fly JAK, that disrupts the pattern of JAK-SAT activation also affects follicle elongation (*Figure 2f,g* and *Figure 2—figure supplement 1e*). Thus, spatial control of JAK-STAT pathway activation is required for follicle elongation. Altogether, these results show that Upd secretion by polar cells and the subsequent gradient of JAK-STAT activation act as developmental cues that define the follicle elongation axis during the early stages of oogenesis.

## MyosinII activity drives apical pulses and early elongation

Once the signal for elongation had been identified, we aimed to determine the molecular motor driving this elongation, which in many morphogenetic contexts is MyosinII (MyoII)(*Heisenberg and Bellaïche, 2013*; *Lecuit et al., 2011*). The knockdown in all FCs of *spaghetti squash* (s*qh*), the MyoII regulatory subunit, leads to a significant decrease in the elongation coefficient and in follicle AR from stage 4 (*Figure 3a–b* and *Figure 3—figure supplement 1b,c*), indicating that MyoII is the motor of early elongation. We have shown that the rotation and the planar polarization of the basal actomyosin is not involved in early elongation. Moreover, at these stages, MyoII is strongly enriched at the apical cortex, suggesting that its main activity is on this domain of the FCs (*Figure 3—figure supplement 1a* and Figure 5c) (*Wang and Riechmann, 2007*). We therefore looked at MyoII on living follicles, focusing on the apical side, and found that it is highly dynamic (*Video 3*). In *Drosophila*, transitory medio-apical recruitment of actomyosin usually drives apical pulses (*Martin et al., 2009*; *Martin and Goldstein, 2014*). Accordingly, using a GFP trap line for Bazooka (Baz-GFP), which concentrates at the *zonula adherens* and marks the periphery of the apical domain, we observed that the transient accumulation of MyoII is associated with a contraction of this domain, which is followed by a relaxation when the MyoII signal decreases (*Figure 3c–e*, *Video 4*). Although we did not find a clear period because cells can pause for a variable time between two contractions, the approximate duration of a pulse was about three minutes. Cross-correlation analysis on many cells from several follicles (n = 86) confirms the association between MyoII and pulses, and reveals that Sqh accumulation slightly precedes the reduction of the apical surface, arguing that it is the motor responsible for these contractions (*Figure 3f*). Inhibiting the activity of Rho kinase (rok), the main regulator of MyoII,

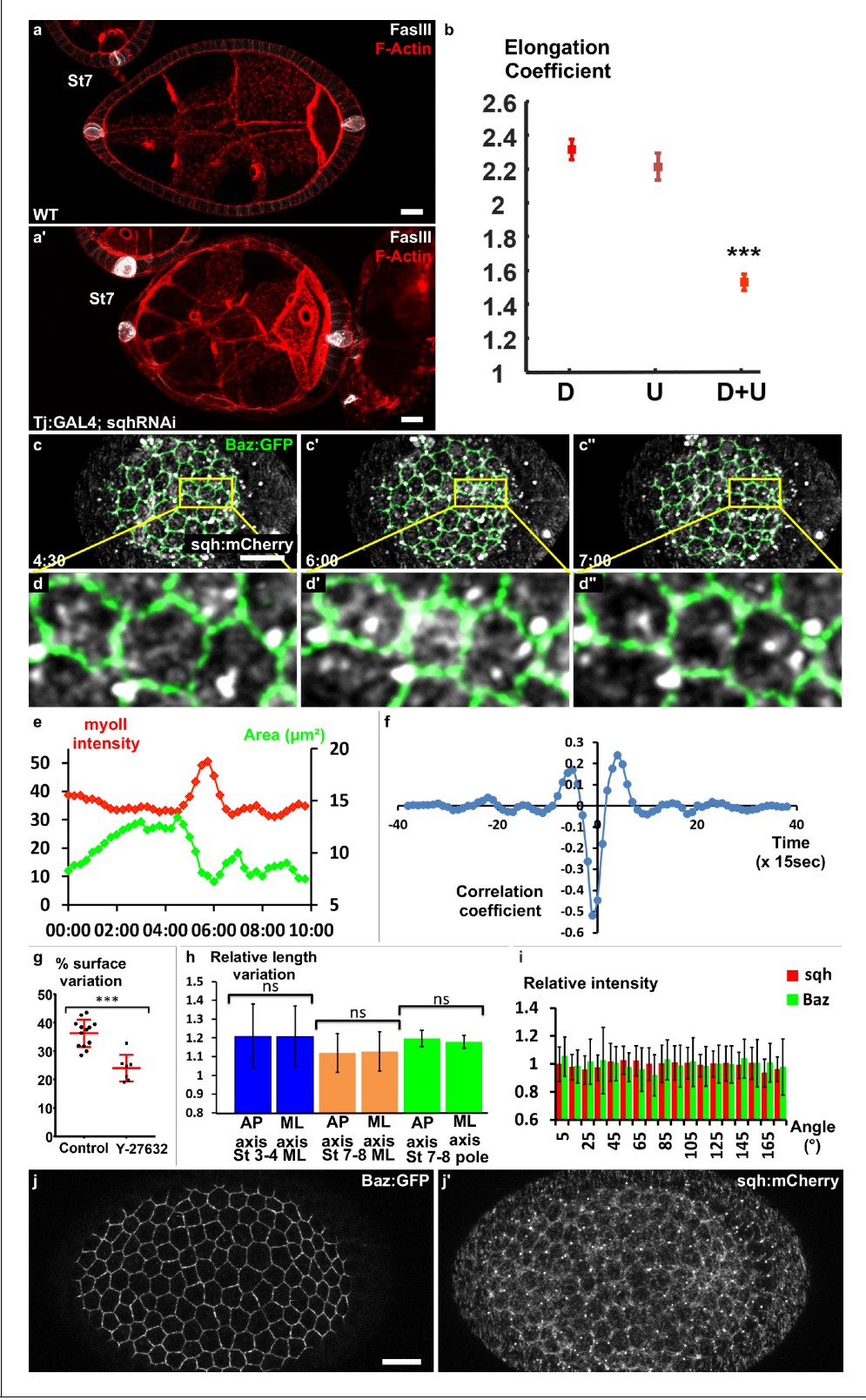

**Figure 3.** Myosin II is required for early elongation and apical pulses. (a) WT and (a') *sqh* knockdown stage 7 follicles stained for F-actin (red) and FasIII (white). (b) Elongation coefficient of WT or *sqh* knockdown follicles during the early elongation phase (D, Driver; U, UAS line) (n > 30). (c) Fluorescence video-microscopy images of a stage 4 WT follicle that expresses BAZ-GFP and Sqh-mCherry. (d) Higher magnification of the area highlighted in (c) showing a pulsing cell. (e) Quantification of the cell apical surface (green) in the cell shown in (d) and of Sqh signal intensity in the apical area (red) over

*Figure 3 continued on next page*

*Figure 3 continued*

time. (**f**) Cross-correlation analysis over time of apical surface and Sqh apical signal intensity based on 86 cells from six follicles at stages 3–4. (**g**) Incubation with the Rok inhibitor Y-27632 strongly reduces pulse activity in stages 3 to 5 WT follicles. Each dot corresponds to one follicle, and at least 10 cells per follicle were analyzed. Red bars represent mean and ± SD (n ≥ 7 follicles). (h) Quantification of the length variation of the follicle cell AP and mediolateral axes during pulses indicates that pulses are isotropic (n > 13 follicles, at least 10 cells per follicle were analyzed). (i) Quantification of the relative BAZ-GFP and Sqh-mCherry signal intensity in cell bounds in function of their angle relative to the AP axis (n = 41 follicles). Relative intensity is given over the mean bond intensity. (j) Fixed stage 7 WT follicle that expresses GFP-Baz and Sqh-mCherry. Scale bars are 10 μm throughout. (p ***<0.001.) Full details of the genotypes and sample sizes are given in the supplementary files.
DOI: https://doi.org/10.7554/eLife.32943.008

The following figure supplement is available for figure 3:

**Figure supplement 1.** Myosin II is required for early elongation.
DOI: https://doi.org/10.7554/eLife.32943.009

using Y-27632, reduces follicle cells' surface variation by ~30% (*Figure 3g*). Thus, MyoII drives apical pulsing during early stages. Consequently, we asked whether and how apical pulses could induce elongation. From stage 9, basal pulses, which are important for the second phase of elongation, have been shown to be anisotropic (*He et al., 2010*). However, quantification of axis length variations showed that the apical pulses were isotropic, both in the mediolateral and polar regions (*Figure 3h*). Tissue elongation is often associated with tissue planar cell polarization, we therefore investigated whether Myosin II and Baz showed exclusive cortical planar polarization, as demonstrated for instance during germband extension (*Bertet et al., 2004*; *Zallen and Wieschaus, 2004*). Consistent with the isotropic nature of the pulses, we failed to detect any oriented enrichment of these proteins, indicating the absence of noticeable apical planar cell polarization of the motor that generates early elongation (*Figure 3i,j*). Altogether, these data indicate that MyoII induces apical pulses and early elongation. Nonetheless, neither the isotropic nature of the pulses nor MyoII localization explains how the pulses could induce elongation.

## JAK-STAT induces a gradient of apical pulses

Our previous results suggest that pulses do not provide an explanation for elongation at a local cellular scale, and we therefore analyzed their spatiotemporal distribution at the tissue scale to determine whether they present a specific tissue pattern. On the basis of the JAK-STAT activity gradient, we hypothesized that cells in the mediolateral part of the follicles should progressively change their behavior during follicle growth. We therefore monitored the mediolateral region of stage 3 and 7 follicles. At stage 3, cells undergo contractions and relaxations asynchronously (*Figure 4a*, *Video 5*). At stage 7, cells were much less active (*Figure 4c*, *Video 6*). This difference was confirmed by monitoring the variation of the relative apical surface of individual cells (*Figure 4e*) or a whole population (*Figure 4f*) (40% of mean variation at stage 3 and only about 15% in the equatorial part at stage 7). Quantification of the average variation of the apical cell surface in a series of follicles

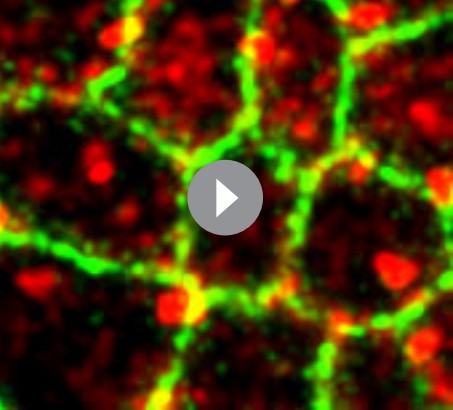

**Video 4.** Zoom in on a cell of a stage 3 follicle expressing Baz-GFP and Sqh-mCherry. Apical MyosinII enrichment occurs at the same time as the apical cell domain contracts.
DOI: https://doi.org/10.7554/eLife.32943.011

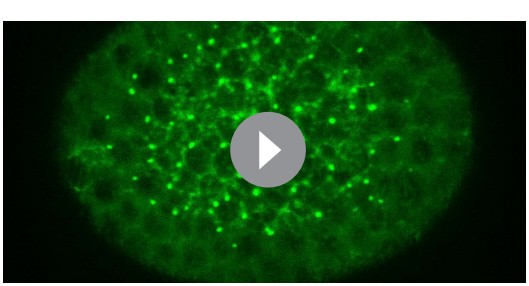

**Video 3.** Stage 3 follicle expressing Sqh-GFP. The pool of apical myosinII is very dynamic.
DOI: https://doi.org/10.7554/eLife.32943.010

indicates that the pulsing amplitude gradually decreases in the mediolateral region from stage 3 to stage 8 (*Figure 4—figure supplement 1a*). This correlation between JAK-STAT activity and pulsing activity in the mediolateral region prompted us to develop a method to visualize the poles of living follicles, which has never been done before (see Materials and methods). We managed to image the poles of stage 3–4 and stage 7–8 follicles, and in both cases the pulse activity is high (*Figure 4b,d–f*, *Videos 7* and *8*). Finally, the analysis of slightly tilted stage 7–8 follicles clearly revealed a gradient of pulse intensity emanating from the pole (*Figure 4g* and *Figure 4—figure supplement 1b*). Thus, the pulse intensity distribution is similar in space and time to the JAK-STAT activity gradient. Moreover, the cell pulse amplitude is significantly reduced in the mediolateral region of stage 3–4 follicles and near the poles of stage 7–8 *upd* RNAi follicles (*Figure 4h,i*, *Videos 9* and *10*), indicating that JAK-STAT regulates FC apical pulsatory activity. Finally, we found that clonal ectopic activation of JAK is sufficient to increase pulse intensity in the mediolateral region of stage 7–8 follicles when compared to similar control clones (*Figure 4j*, *Videos 11* and *12*). Together, these results show that the JAK-STAT pathway has an instructive role in controlling the intensity of FC apical pulses, leading to a specific spatiotemporal pattern breaking follicle symmetry in each hemisphere.

## Myosin II is required at the poles but is not controlled by JAK-STAT

Since the JAK-STAT pathway and MyoII are both important for apical pulses, we studied their functional relationship. The apical level of the Myosin II active form, visualized by its phosphorylation, is significantly reduced by 18% in *STAT92E* null mutant clones on young follicles when compared to WT surrounding cells (n = 17 clones, p<0,001), which may suggest that MyoII activity is regulated by JAK-STAT signaling (*Figure 5a*). However, clonal gain of function of JAK in the region where the JAK-STAT pathway is normally inactive (mediolateral at stage 7–8) does not increase the apical phosphorylation level of MyoII (*Figure 5b*). Moreover, analysis of the global pattern of apical MyoII phosphorylation does not reveal any gradient between the poles and the mediolateral regions (*Figure 5c,d*). Altogether, these data indicate that MyoII activation by phosphorylation is independent of JAK-STAT signaling and that JAK-STAT regulates pulses by another means, which might be required for efficient apical recruitment of MyoII. Thus, although JAK-STAT and Myosin II are both required for early elongation, they control pulses in parallel.

If the gradient of apical pulses induces early elongation and explains MyoII involvement in this process, then MyoII function should be required at the poles. We generated mutant clones for a null allele of *sqh* to analyze where MyoII is required for elongation. As previously shown (*Wang and Riechmann, 2007*), such clones reach a limited size, probably explaining why it is rare to obtain a clone that covers poles, especially after stage 5. We focused on clones covering the anterior pole. To quantify the effect of mutant clones on semi-follicles, we measured extrapolated Aspect Ratio (eAR) of each semi-follicle, which means, the ratio of the corresponding full ellipse (see Materials and methods and *Figure 5e*). For a WT follicle, the anterior eAR is equal or superior to the posterior eAR, as the anterior pole is normally more pointed than the posterior (*Figure 5g*). Analysis of the eAR of the poles containing such mutant clones indicates that Myosin II loss of function specifically affects the elongation of this pole, compared to the opposite WT posterior poles (n = 10) (*Figure 5f,i*). Moreover, we never observed clones in the mediolateral regions inducing elongation defects (n = 35) (*Figure 5h*). Finally, we also performed similar experiment with *Rok* null mutant clones. Such clones have a weaker effect on cell morphology (*Figure 5j* and *Wang and Riechmann, 2007*), but still affect elongation when situated at the pole (*Figure 5f,j*). Thus, MyoII and Rok are required specifically at the poles to induce early elongation. These results strongly argue that the gradient of apical isotropic FC pulses is the force-generating mechanism that drives early elongation.

## Early elongation is associated with cell constriction and cell intercalation

Independently of the upstream events, we asked which cellular behavior was associated with early elongation. The simplest possibility would be that cells are stretched along the AP axis. However, cells are actually slightly elongated perpendicularly to the axis of elongation and this morphology did not change significantly over time, indicating that this parameter does not contribute to follicle elongation during early stages (*Figure 6—figure supplement 1a,b*). Tissue elongation can be also

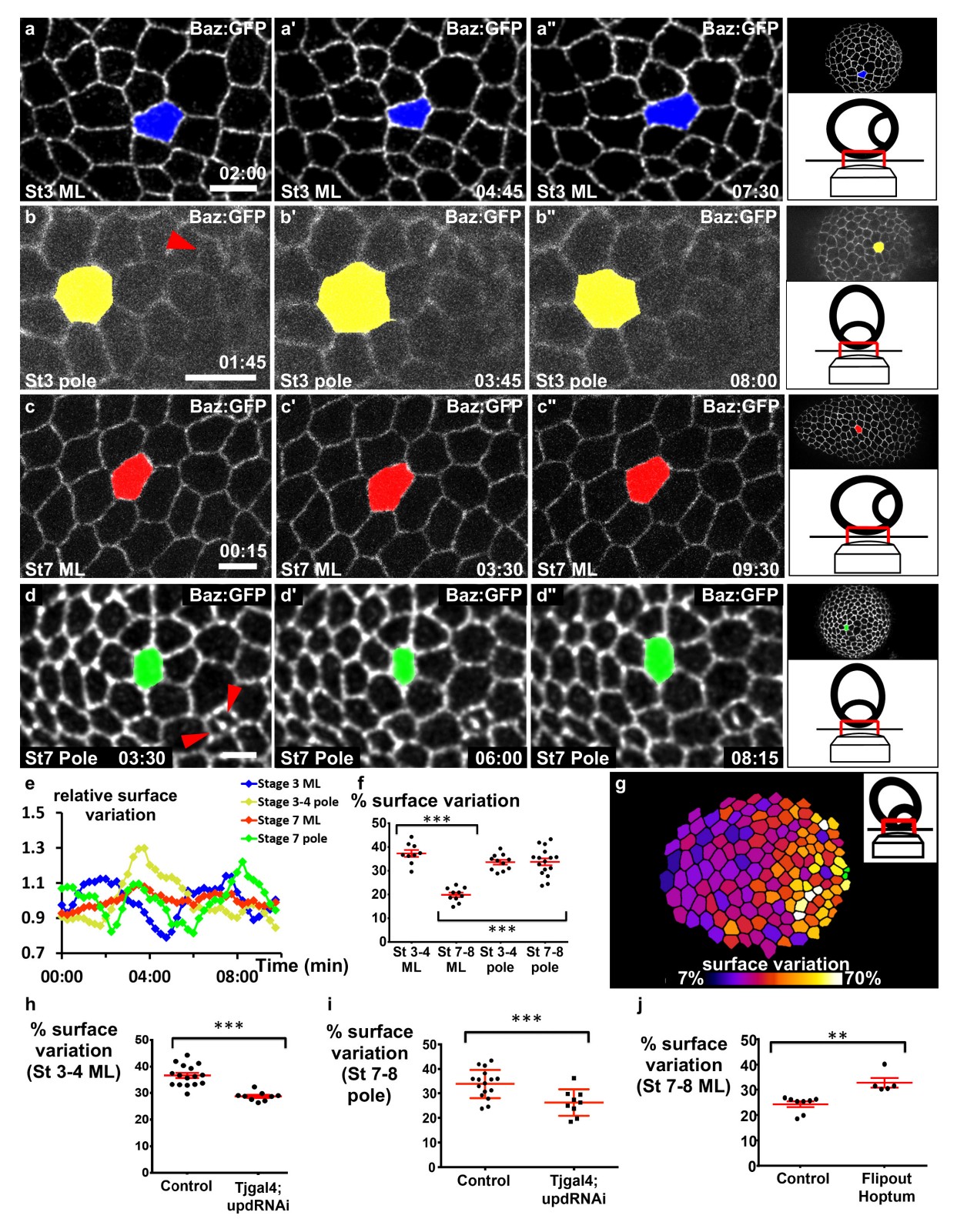

**Figure 4.** JAK-STAT induces a double gradient of pulses. (**a–d**) Images from movies of the mediolateral region of (**a**) stage 3 and (**c**) stage 7 BAZ-GFP expressing follicles, or of the area near the polar cells (red arrowheads) of (**b**) stage 3 and (**d**) stage 7 follicles. Scale bars: 10 μm. (**e**) Surface variation of individual cells (examples shown in [a–d]) as a function of time (ML, mediolateral). The surface of each cell is divided by its average surface over time. (**f**) Mean percentage of apical surface variation depending on stage and position (n ≥ 9 follicles). (**g**) Color-coding of the pulse intensity of a representative

*Figure 4 continued on next page*

*Figure 4 continued*

stage 7 follicle (tilted view from the pole, see schematic image in insert) reveals an intensity gradient from the polar cells (in green) to the mediolateral region. (h–j) Mean percentage of apical surface variation in the mediolateral region of (h) stage 3–5 follicles and (j) stage 7 to 8 follicles, and (i) at the pole of stage 7–8 follicles for the indicated genotypes. (h, i) n ≥ 9, (j) n ≥ 5. (p **<0.01, ***<0.001, red bars represent mean and ± SD). Full details of the genotypes and sample sizes are given in the supplementary files.

DOI: https://doi.org/10.7554/eLife.32943.012

The following figure supplement is available for figure 4:

**Figure supplement 1.** Myosin II is required for early elongation.

DOI: https://doi.org/10.7554/eLife.32943.013

associated with oriented cell divisions. A movie of mitosis in the FE showed that this orientation is really variable through the different steps of mitosis (*Figure 6—figure supplement 1c*). We therefore quantified the orientation of cytokinesis figures, which did not highlight any bias towards the AP axis (*Figure 6—figure supplement 1d*). Finally, we asked whether early elongation could be associated with cell intercalation. Analysis of fluorescence video-microscopy images gave inconclusive results because such events are probably rare and slow, and because follicle rotation precludes their reproducible observation (*Video 13*). We therefore used an indirect method. As follicle cells from stage 6 onwards stop dividing and their number remains constant, we counted the number of cells in the longest line of the AP axis (i.e., the follicle plane that includes the polar cells). This number significantly increases between stage 6 and 8, showing that cells intercalate in this line (*Figure 6a–d*). This number is also correlated with the follicle AR (*Figure 6e*), indicating that follicle early elongation is associated with cell intercalation along the AP axis. Cell intercalation can be powered at a cellular scale by the polarized enrichment of Myosin II in the cells that rearrange their junctions (*Bertet et al., 2004*). However, we have already shown that MyoII does not show such a pattern in FCs (*Figure 3i,j*). Alternatively, intercalation can be promoted at a tissue scale. For instance, apical cell constriction in the wing hinge induces cell intercalations in the pupal wing (*Aigouy et al., 2010*). We observed that the cell apical surface is lower at the poles than in more equatorial cells, and that this difference increases during the early elongation phase (*Figure 6f,g,h*). Such a difference could be explained by cell shape changes or by a differential cell growth. Cell height is significantly larger at the poles, indicating that the changes in apical surface are linked to cell morphology, as previously shown during mesoderm invagination for instance (*Figure 6i*) (*He et al., 2014*). However, cells at the poles have a lower volume than those in the mediolateral region at stage 7 (*Figure 6—figure supplement 1e*). This difference of volume is nonetheless proportionally weaker than the change in apical surface, suggesting the cell shape changes induce the reduction of volume rather than the opposite. Thus, early elongation is associated with a moderate cell constriction in the polar regions. *sqh* mutant FCs are stretched by the tension coming from germline growth, a defect opposite to cell constriction (*Figure 5g,h*) (*Wang and Riechmann, 2007*). Interestingly, FCs that are mutant for *Stat92E* are also flattened, with a larger surface and a lower height, compared to WT surrounding cells (*Figure 6j–m*). Moreover, the apical cell surface at the

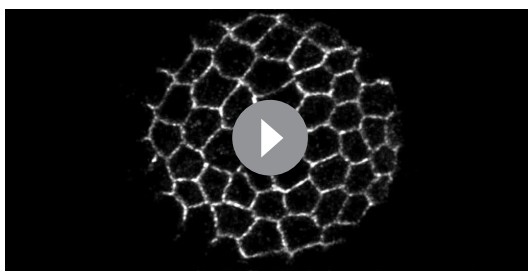

**Video 5.** Stage 3 follicle expressing Baz-GFP. Cells in the mediolateral part undergo apical pulsations.
DOI: https://doi.org/10.7554/eLife.32943.014

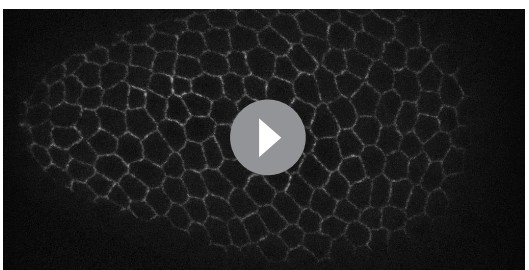

**Video 6.** Stage 7 follicle expressing Baz-GFP. The apical surface variation is strongly reduced on the mediolateral part compared with stage 3 follicles.
DOI: https://doi.org/10.7554/eLife.32943.015

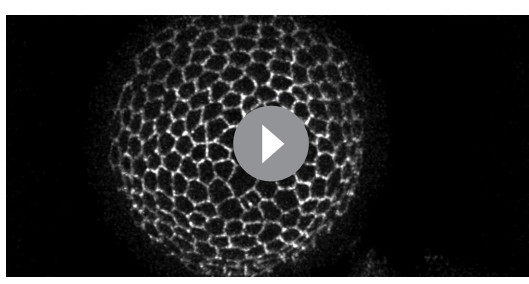

**Video 7.** Stage 7 follicle expressing Baz-GFP observed from the pole. Polar cells are indicated on the corresponding *Figure 5d* (red arrowheads). The pulse intensity remains high in these cells compared with *Video 4*. The rotation is visible and occurs around the polar cells.

DOI: https://doi.org/10.7554/eLife.32943.016

poles of stages 7–8 is increased by the loss of function of Upd (*Figure 6h*). Hence, these results link JAK-STAT and the morphology of the follicle cells in a coherent manner with an involvement of apical pulses for the cell constriction observed at the poles.

Altogether these results indicate that two cell behaviors occur during the early phase of elongation: oriented cell intercalation towards the A-P axis and apical cell constriction at the poles.

## Discussion

The first main conclusion of this work is that follicle elongation can be subdivided into at least two main temporal and mechanistic phases: an early one (stages 3–6) that is independent of Fat2, rotation, and ECM and F-actin basal polarization, and a second one (stages 7–14) that requires Fat2. This is reminiscent of germband extension where different elongation mechanisms have been described (*Lye et al., 2015*; *Collinet et al., 2015*; *Rauzi et al., 2010*; *Blankenship et al., 2006*; *Sun et al., 2017*). In the case of the follicle, it is still not clear how overlapping and interconnected these different mechanisms are.

Fat2 has no role in early elongation. Nevertheless, Fat2 is required as early as the germarium for the correct planar polarization of the microtubule cytoskeleton and for follicle rotation, which takes place during the early elongation phase (*Viktorinová and Dahmann, 2013*; *Chen et al., 2016*). The rotation reinforces the basal pcp of the F-actin during stages 4–6, and thus probably participates in the late phase in this way (*Cetera et al., 2014*; *Aurich and Dahmann, 2016*). Rotation is also necessary for the deposition of ECM fibrils, although their specific role in elongation has not been clearly elucidated yet. Another mechanism that participates in elongation is the ECM stiffness gradient (*Crest et al., 2017*). However, its contribution begins only at stage 7–8. This is in agreement with the fact that the ECM stiffness gradient depends on Fat2 and that *vkg* (ColIV) loss-of-function follicles elongate correctly up to stage 8, showing that the ECM is required only in the second elongation phase (*Crest et al., 2017*; *Haigo and Bilder, 2011*). Thus, the setting up of the elements required for this second elongation phase fully overlaps with the first elongation phase, but these two phases are so far unrelated at the mechanistic level. Notably, the early elongation phase requires elements of the apical side of follicle cells, whereas the second phase involves the basal side. Mirroring our observations, a recent report nicely shows that the fly germband extension, which was thought to depend exclusively on the apical domain of the cells, also involves their basal domain (*Sun et al., 2017*). As both Fat2 and the gradient of BM stiffness are involved in the elongation at

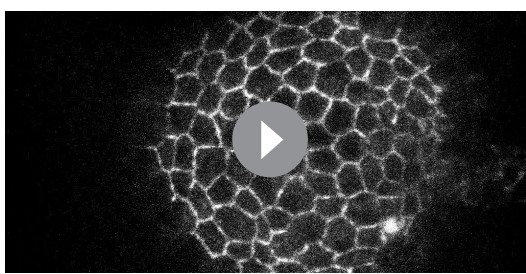

**Video 8.** Stage 3 follicle expressing Baz-GFP observed from the pole. Polar cells are indicated on the corresponding *Figure 5b* (red arrowhead).
DOI: https://doi.org/10.7554/eLife.32943.017

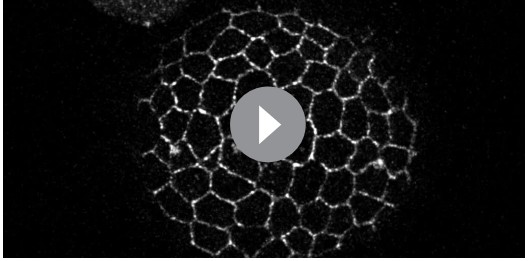

**Video 9.** Stage 3 *upd* knockdown follicle expressing Baz-GFP. The intensity of the pulse is reduced compared with a WT stage 3 follicle (*Video 3*).
DOI: https://doi.org/10.7554/eLife.32943.018

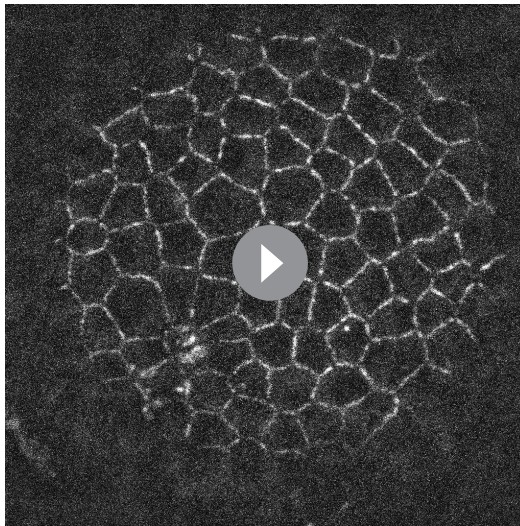

**Video 10.** Stage 7 *upd* knockdown follicle expressing Baz-GFP and observed from the pole. The intensity of the pulse is reduced compared with a WT stage 7 follicle (*Video 6*).
DOI: https://doi.org/10.7554/eLife.32943.019

stage 8 and as apical pulses are still observed at this stage, it appears that the apical and basal domain contributions may slightly overlap. Moreover, both the gradients of apical pulses and of BM stiffness are under the control of JAK-STAT, indicating that this pathway has a pleiotropic effect on follicle elongation.

We have also shown that integrin and Pak contribute to early elongation in an indirect manner through their impact on the positioning, the differentiation or the survival of the polar cells. In this respect, *Pak* and *mys* mutants belong to a new phenotypic class that could also comprise the Laminin β1 subunit (LanB1) and the receptor-like tyrosine phosphatase Lar (*Díaz de la Loza et al., 2017*; *Frydman and Spradling, 2001*). We do not yet know how the A-P position of those cells is established and maintained. Interestingly, *Pak* mutants also have an altered germarium structure leading to abnormal follicle budding, suggesting that polar cell mispositioning might be linked to this primary defect (*Vlachos et al., 2015*). However, it is worth noticing that *Pak* mutant follicles do not elongate at all, whereas they still have a cluster of polar cells. Thus, *Pak* might also be required for early elongation in a more direct manner than polar cell positioning, downstream of or in parallel to the JAK-STAT pathway, but independently of basal planar polarization.

We found that polar cells define the elongation axis of each follicle during early elongation by secreting the Upd morphogen and by forming a gradient from each pole, which in turn induces apical pulses. The isotropic nature of these pulses does not provide an evident link with tissue elongation, unlike the oriented basal pulses going on in later stages (*He et al., 2010*). Moreover, the absence of planar polarization of MyoII in apical regions, which is the driving force of early elongation, and the non-requirement for 'basal pcp' strongly argue against a control of this elongation phase by a planar cell polarity working at a local scale. Instead, several strong arguments propose that early elongation relies on pulses working at a tissue scale (*Figure 6n*). First, the pulses are distributed in a gradient from the poles, suggesting that this distribution can orient the elongation in each hemisphere. Also, our data indicate that JAK-STAT does not directly regulate MyoII activity, and, thus, that they probably work in parallel to control pulses. The convergence of requirement for JAK-STAT and myosin II activities for both pulses and early elongation argues for a causal link between these two processes. To date, JAK-STAT has no other known morphogenetic function

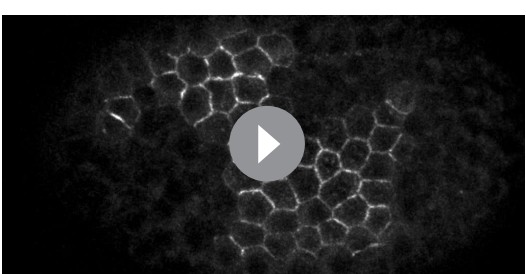

**Video 11.** Stage 7 follicle expressing ectopically Baz-mCherry. The intensity of the pulse is low.
DOI: https://doi.org/10.7554/eLife.32943.020

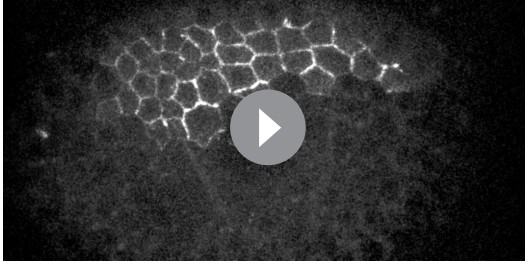

**Video 12.** Stage 7 follicle expressing ectopically Baz-mCherry and Hop[tum]. Activation of the JAK-STAT pathway is sufficient to increase the pulsing.
DOI: https://doi.org/10.7554/eLife.32943.021

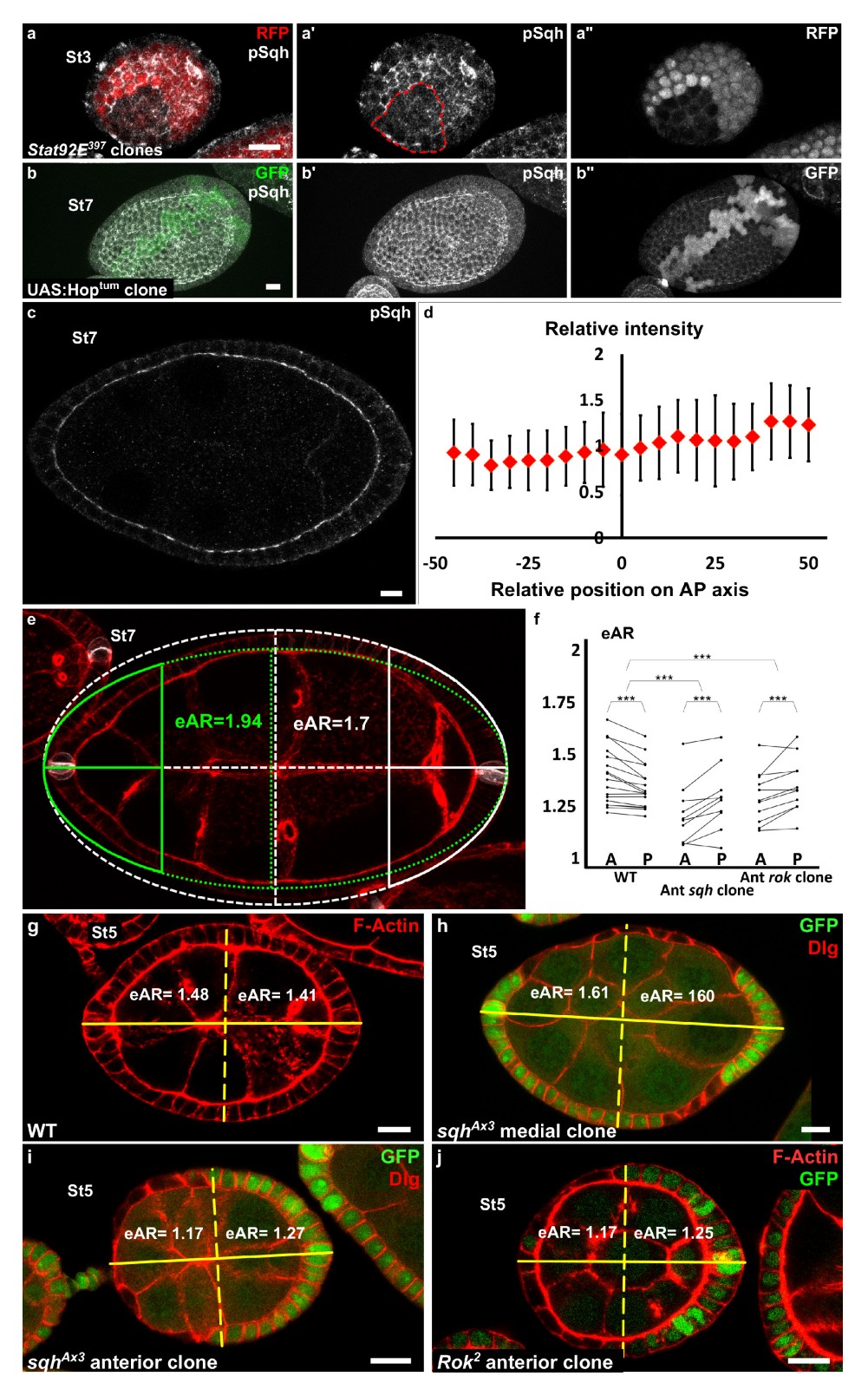

**Figure 5.** Myosin II is not controlled by JAK-STAT but is required at the poles. (a) Apical level of phosphorylated Sqh (pSqh, white and [a']) is reduced in a mutant *Stat92E* clone (RFP-negative) in a stage 3 follicle (z-projection of the superior half of the follicle). (b) Clonal overexpression of *Hop^tum* (green cells) on a stage 7 follicle is not sufficient to increase the expression of apical pSqh (white and [b']) z-projection of the superior half of the follicle). (c) pSqh staining in the middle plane of a wildtype stage 7 follicle. (d) Quantification of the intensity of apical pSqh along the AP axis of stage 6–7 follicles.
*Figure 5 continued on next page*

*Figure 5 continued*

n = 5 follicles. Baseline value = mean apical pSqh per follicle. (**e**) Illustration of extrapolated Aspect Ratio (eAR) calculation based on width measure of a pole at 25% of AP axis length. (**f**) Quantification of the extrapolated aspect ratio (eAR) of stage 4–7 WT follicles or follicles with a $sqh^{AX3}$ or $Rok^2$ clone covering the anterior pole (n ≥ 10). In WT follicles, the anterior is significantly more curved than the posterior, whereas the tendency is opposite with *sqh* and *Rok* clones. (p ***<0.001.) (**g,h,I,j**) representative images of (**g**) WT, (**h**) mediolateral $sqh^{AX3}$ clone, (**i**) anterior $sqh^{AX3}$ clone and (**j**) anterior $Rok^2$ clone with the corresponding eARs. Full details of the genotypes and sample sizes are given in the supplementary files.
DOI: https://doi.org/10.7554/eLife.32943.022

before stage 8. Similarly, the only other known function of MyoII is linked to the rotation, which is not involved in early elongation, and MyoII is very concentrated at the apical cortex, emphasizing the role of this domain. Moreover, though present all around the follicle, MyoII is required for early elongation at the poles. Thus, the apical localization and the spatiotemporal requirement of MyoII are coherent with the action of apical pulses as the driving force for early elongation.

JAK-STAT has already been implicated in the elongation of different tissues in flies and in vertebrates. For instance, Upd works as the elongation cue for the hindgut during fly embryogenesis, a process also associated with cell intercalation, although the underlying mechanism is unknown (*Johansen et al., 2003*). Maybe more significantly, JAK-STAT is involved in the extension-convergence mechanism during zebrafish gastrulation (*Yamashita et al., 2002*). Moreover, JAK-STAT also participates in other morphogenetic events, such as tissue folding in the fly gut and wing disc (*Wells et al., 2013*). All these roles are potentially linked to a control of apical cell pulses. As our results indicate that this control is not through MyoII activation, identifying the transcriptional targets of STAT that explain its impact on apical actomyosin will be relevant for many developmental contexts.

How the apical pulses precisely drive early elongation remains a question that will require further investigations. Nonetheless, we determined that early elongation is associated with apical cell constriction close to the poles and oriented cell intercalations. Cell constriction is probably a direct consequence of apical pulses, as has been shown in many other contexts, because both myosin II and JAK-STAT loss of function affect pulse and induce an increase of the apical surface (*Wang and Riechmann, 2007*; *Martin and Goldstein, 2014*). Thus, as during tissue invagination, cell constriction may accentuate the curvature at the poles and thus promote elongation. Intercalation can be induced at a tissue scale by long-range anisotropic tensions in the tissue, as exemplified by the development of pupal wings or mammalian limb bud ectoderm (*Aigouy et al., 2010*; *Lau et al., 2015*). In the wing, elongation is due to contraction of the hinge, which corresponds to an apical constriction of the cells. Here, the apical pulses could act in a similar way via the constriction, acting as a pulling force at each pole. Thus, intercalations may correspond to a passive response, bringing plasticity to the tissue and hence stabilizing its elongation. Although the respective contribution of these two cell behaviors - apical constriction at the poles and cell intercalation along the AP axis – and their potential links remain to be determined, together they probably recapitulate at the cellular scale the elongation observed at the tissue scale. Importantly, such a mechanism does not require any planar cell polarization, in agreement with our observations. A gradient of randomly oriented cell migration contributes to vertebrate AP axis elongation and is, to our knowledge, the only other example of a tissue elongation mechanism instructed by a signaling cue and independent of pcp (*Bénazéraf et al., 2010*), in contrast to the many examples where pcp controls cell-movements that induce axis elongation in vertebrates. Our work proposes an alternative mechanism to explain how a morphogen gradient can induce elongation solely through transcription activation, and without any requirement for a polarization of receiving cells. This simple mechanism may apply to other tissues and other morphogens.

## Materials and methods

### Genetics

All the fly stocks with their origin and reference are described in *Supplementary file 1*. The detailed genotypes, temperature and heat-shock conditions are given in *Supplementary file 2*.

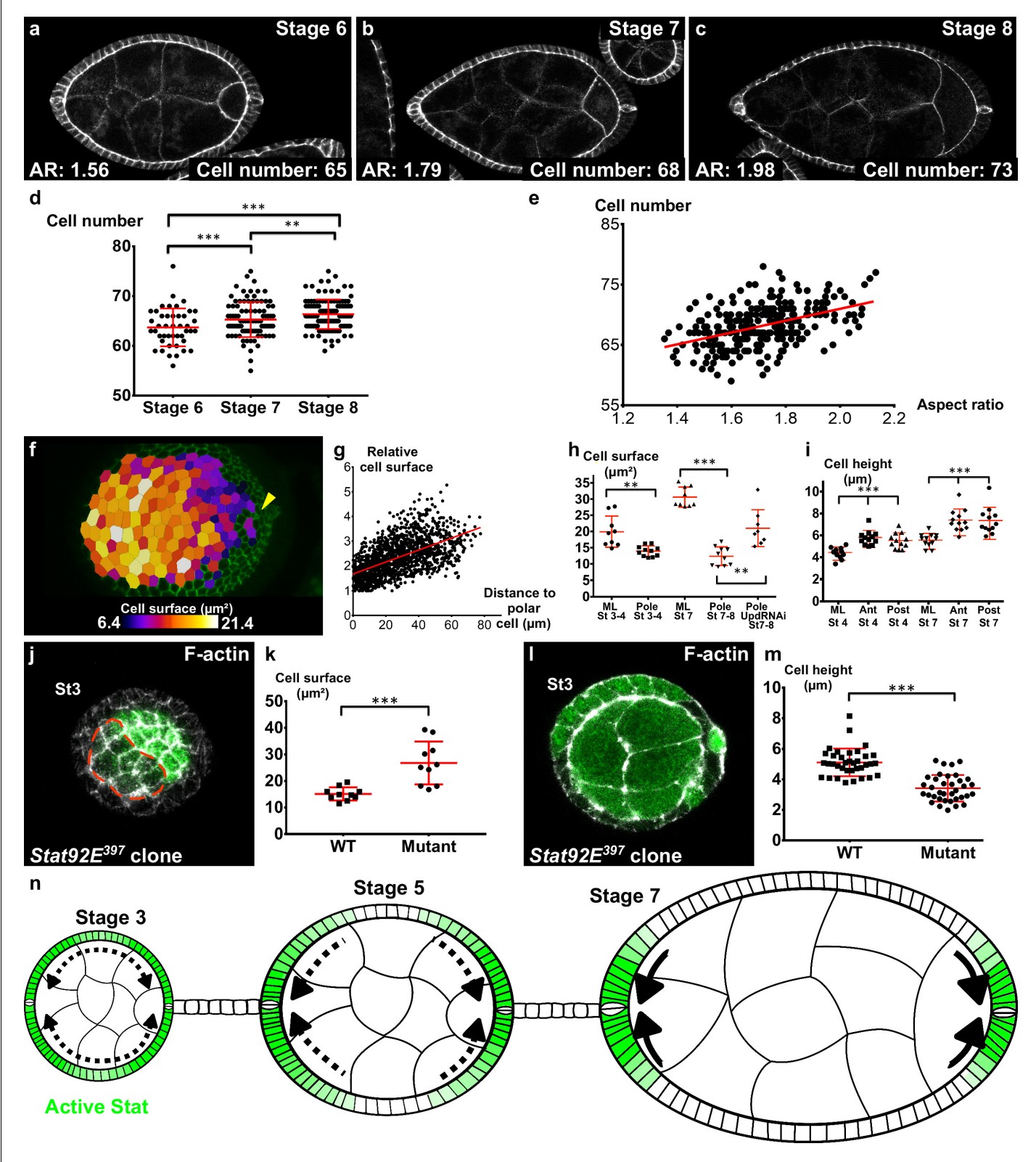

**Figure 6.** Localized apical cell constriction and oriented cell intercalation occur during early elongation. (a–c) AR and number of cells along the polar cell plane in representative (a) stage 6, (b) stage 7 and (c) stage 8 follicles stained for DE-Cad. (d–e) Number of follicular cells in the plane of polar cells based on DE-Cad staining of stage 6–8 follicles depending on (d) the stage and (e) the aspect ratio of the follicle. (f) Heat map of the cell apical surface of a representative stage 7 follicle imaged as on *Figure 4g*. Arrowhead shows polar cells. (g) Quantification of the relative apical cell surface (smallest

*Figure 6 continued on next page*

*Figure 6 continued*

cell = 1) as a function of the distance from polar cells (n = 10 stage 7 follicles, 1487 cells). (h) Apical cell surface and (i) cell height depending on stage, position and genotype (ML, mediolateral). (j) Representative top view and (l) section view of *Stat92E³⁹⁷* mutant clones at stage 3. Mutant cells have (k) a larger apical surface and (m) a lower cell height than do wildtype cells. Each dot corresponds to the mean value obtained for a clone. (n) Schematic figure showing the progressive restriction of JAK-STAT signaling (green) and of cell constriction to the follicle poles. p *<0.05, **<0.01, ***<0.001. Full details of the genotypes and sample sizes are given in the supplementary files.

DOI: https://doi.org/10.7554/eLife.32943.023

The following figure supplement is available for figure 6:

**Figure supplement 1.** No contribution of cell elongation and cell division to early elongation.

DOI: https://doi.org/10.7554/eLife.32943.024

## Immunostaining and imaging

Dissection and immunostaining were performed as described previously (*Vachias et al., 2014*) with the following exceptions: ovaries were dissected in Supplemented Schneider, each of the ovarioles was separated before fixation to obtain undistorted follicles. Primary antibodies were against pMyoII (1/100, Cell Signaling #3675), DE-Cad (1/100, DHSB #DCAD2), Dlg (1/200 DHSB #4F3), and FasIII (1/200, DHSB #7G10). Images were taken using a Leica SP5 or SP8 confocal microscope. Stage determination was performed using unambiguous reference criteria, which are independent of follicle shape (*Spradling, 1993*).

For live imaging, ovaries were dissected as described previously (*Prasad et al., 2007*) with the following exceptions: each ovariole was separated on a microscope slide in a drop of medium and transferred into a micro-well (Ibidi BioValey) with a final insulin concentration of 20 µg/ml. Samples were cultured for less than 2 hr before imaging with a Leica SP8 confocal using a resonant scanner. Follicles were incubated with Y-27632 (Sigma) (diluted in PBS to 250 µM) for 10–30 min before image acquisition. To image the poles, glass beads were added into the well to form a monolayer (Sigma-Aldrich, G4649 for stage 6–8 or G1145 for earlier stages). Ovarioles were added on top of the beads and follicles falling vertically between the beads were imaged.

Cell pulse analysis was performed using the Imaris software and a MATLAB homemade script to segment and measure the cell surface on maximum intensity projections of 40 stacks taken every 15 s. The intensity of one cell pulsation corresponds to: (maximum surface of the cell – min surface)/ (mean surface). The isotropy of one cell pulse is measured by dividing the AP and ML bounding box (best fit rectangle) axis length at cell maximal area by the AP and MP bounding box axis length, respectively, at the cell's minimal area. For each follicle, at least 10 cells were analyzed. For visualization (images presented in *Figure 4a,c,d* and the attached movies), the original files were deconvolved, but all the analyses were carried out using the raw files.

The Fiji software was used to measure the length of the long and short axis of each follicle on the transmitted light channel, and then to determine the aspect ratio in WT and mutant follicles. Cells in the longest line of the AP axis were counted manually using Fiji on the DNA and DE-Cadherin channels. Bazooka-GFP and MyosinII-mCherry enrichment were analyzed using the Packing Analyser software (*Aigouy et al., 2010*). Cells were semi-automatically segmented on the basis of the Baz-GFP channel that was used as common pattern to calculate the intensity of each bond for both channels.

Fiji was used to measure the intensity of the pSqh signal and the 10XStatGFP signal. A 15-pixel wide line was drawn using the freehand tool, either within the cells (10X StatGFP) or at the apical level of the cells (pSqh), from the anterior to the posterior of cross-sectional images of follicles.

The extrapolated aspect ratio (eAR) was estimated for each pole by measuring the width of the follicle at 25% of its total length: for any given

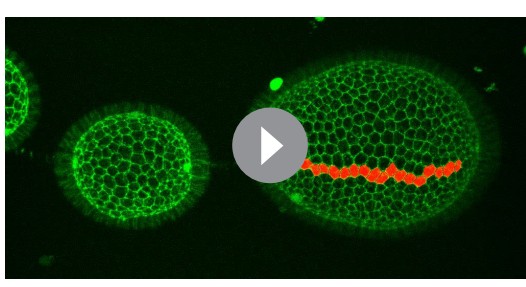

**Video 13.** Movie representing a stage 7 DE-Cad-GFP follicle imaged over two hours. One cell is tracked (red line). No intercalation occurs during this period.
DOI: https://doi.org/10.7554/eLife.32943.025

ellipse, this value corresponds to $\sqrt{3}/2$ times its total width. Therefore, this measure allows the extrapolation of a width and an aspect ratio for each pole. Follicles with gaps in the epithelium were excluded on the basis of Dlg staining.

To measure cell elongation, images of DE-Cadherin-GFP-expressing follicles were semi-automatically segmented using the Packing Analyser software, and for each follicle, the elongation tensor was calculated. The elongation tensor was defined by the mean elongation of all the segmented cells (elongation magnitude) and the mean orientation.

The rose diagrams were generated with Packing Analyser; each bin represents a 10° range and the bin size is proportional to the number of acquired data. Cell volume was obtained by the multiplication of the mean surface and the mean height of the cells.

Figures were assembled using ScientiFig (*Aigouy and Mirouse, 2013*).

## Statistical analysis

For all experiments, sample size is indicated in the figure legends or in *Supplementary file 3*. No statistical method was used to predetermine sample size. Results were obtained from at least two independent experiments, and for each experiment, multiple females were dissected. No randomization or blinding was performed. For each experimental condition, variance was low. Matlab software has been used to perform analysis of covariance to determine the elongation coefficient, and multiple pairwise comparison tests were run to determine the p-value between different conditions (*aoctool* and *multicompare*, Statistic and Machine Learning Toolbox). The normality of the samples was calculated using a D'Agostino and Pearson normality test. The unpaired t-test was used to compare samples that had a normal distribution. The unpaired Mann-Whitney test was used to compare samples that were not normally distributed. For comparison of eAR of anterior and posterior poles, a two-way ANOVA test with repeated measures was conducted on both poles and for two genotypes. The post-hoc analysis (two pair-wise Bonferroni tests) was performed. When shown, error bars represent SD. For all figures, p *<0.01, **<0.005, ***<0.001.

## Acknowledgements

We thank R Basto, M Crozatier, C Dahmann, M Grammont, D Harrison, A-M Pret and E Wieschaus for fly stocks or reagents. This work was funded by the ATIP-Avenir program, Association pour la Recherche contre le Cancer (ARC) and the Auvergne Region. We also thank the confocal imaging facility of Clermont-Ferrand (ICCF) and team members for comments on the manuscript.

## Additional information

### Funding

| Funder | Grant reference number | Author |
|---|---|---|
| Fondation ARC pour la Recherche sur le Cancer | ATIP-Avenir | Vincent Mirouse |
| Institut National de la Santé et de la Recherche Médicale | ATIP-Avenir | Vincent Mirouse |

The funders had no role in study design, data collection and interpretation, or the decision to submit the work for publication.

### Author contributions

Hervé Alégot, Conceptualization, Data curation, Formal analysis, Supervision, Funding acquisition, Validation, Investigation, Methodology, Writing—original draft, Project administration, Writing—review and editing; Pierre Pouchin, Conceptualization, Resources, Data curation, Formal analysis, Validation, Investigation, Visualization, Methodology, Writing—original draft, Writing—review and editing; Olivier Bardot, Resources, Data curation, Formal analysis, Validation, Visualization, Methodology; Vincent Mirouse, Conceptualization, Formal analysis, Supervision, Funding acquisition, Investigation, Methodology, Writing—original draft, Project administration, Writing—review and editing

**Author ORCIDs**
Pierre Pouchin (iD) http://orcid.org/0000-0003-3858-3152
Vincent Mirouse (iD) http://orcid.org/0000-0001-5823-342X

**Decision letter and Author response**
Decision letter https://doi.org/10.7554/eLife.32943.033
Author response https://doi.org/10.7554/eLife.32943.034

## Additional files

**Supplementary files**
• Supplementary file 1. Stock list and source. The Baz-Cherry fusion protein was produced by cloning mCherry in frame at the C-terminus of the Par-6 coding sequence in the pUASP vector.
DOI: https://doi.org/10.7554/eLife.32943.026

• Supplementary file 2. Detailed genotypes and specific conditions. HS: 1 hr heat-shock at 37°C, when not specified flies were kept at 25°C.
DOI: https://doi.org/10.7554/eLife.32943.027

• Supplementary file 3. Detailed sample size. n corresponds to the number of analyzed follicles, with usually more than 10 segmented cells per follicle.
DOI: https://doi.org/10.7554/eLife.32943.028

• Transparent reporting form
DOI: https://doi.org/10.7554/eLife.32943.029

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
