## [Decision Letter]

[Editors’ note: a previous version of this study was rejected after peer review, but the authors submitted for reconsideration. The first decision letter after peer review is shown below.]

Thank you for submitting your work entitled "Jak-Stat pathway induces *Drosophila* follicle elongation by a gradient of apical contractility" for consideration by *eLife*. Your article has been reviewed by two peer reviewers, and the evaluation has been overseen by a Reviewing Editor and a Senior Editor. The reviewers have opted to remain anonymous.

Our decision has been reached after consultation between the reviewers. Based on these discussions and the individual reviews below, we regret to inform you that your work will not be considered further for publication in *eLife*.

Both reviewers agreed that there is a lot of new, important information contained within the manuscript, particularly describing the new process of elongation at early stages of oogenesis, but that there is definitely more work to do to make the data presented more solid. We would welcome a substantially revised paper in the future, but the authors should address all of the reviewers' concerns before beginning a new submission.

Additionally:

In the Abstract the authors write that elongation occurs "without planar cell polarity requirement". However, they showed lack of planar polarization just for MyoII and Baz (subsection “MyosinII activity drives apical pulses and early elongation”).

Subsection “Polar cells define the axis of early elongation”, second paragraph: In Pak mutants, the authors often observe a single polar cell cluster (it should be Figure 1, not 1D) and conclude, that Pak is required for polar cell "positioning". If it is just involved in positioning, I would expect two follicle cell clusters in each case. For me it looks that Pak is required either for polar cell specification or for polar cell survival.

I also had some difficulties to find the link between the JAK-STAT activity gradient and the morphogenetic behavior (intercalation, stretching) described in the second part.

*Reviewer #1:*

The manuscript by Alegot et al. focuses the elongation of the *Drosophila* follicle. Previous work established that this morphogenesis depends on a whole-tissue rotation; however, this paper reports that the follicle completes the earliest phase of elongation when rotation is blocked. The authors use two conditions in which the follicle remains spherical (loss of Pak and Mys) to show that the early elongation defect correlates with defects in the positioning of the polar cells. They then perform several experiments that suggest that an Upd signal from the polar cells is required for early elongation. Finally, they present data and propose a model suggesting that a gradient of myosin-based apical contraction emanating from the poles causes a convergence and extension type process in the epithelium.

The ovarian follicle has emerged as a powerful system to identify and study novel mechanisms contributing to tissue elongation. As such, the introduction of a new mechanism that feeds into this process represents an important contribution to this field. Moreover, if the follicle cells really are intercalating with one another in a directional manner in the absence of a planar polarized cue, this observation is likely to be of broad interest to the morphogenesis community. As it stands, however, I have concerns about some of the experiments and their interpretations.

Statistical analyses need to be reported for the graph in Figure 1. This will likely require increasing the n for many of the *fat2* measurements as some of them are currently very low (including one stage that is zero).

Figure 1 report aspect ratios using a range of stages (4-8). The range is problematic. If one group has more stage 8 follicles and the other has more stage 4 follicles, it is possible to obtain a false positive result. These types of comparisons can only be made when all of the follicles are at a single stage.

Figure 1 purports to show that the polar cells are mis-positioned because they are not adjacent to the stalk. However, the stalk moves away from the polar cells during mid-oogenesis, a phenomenon that can be seen in the oldest follicle in Figure 1. The oocyte appears to be mis-localized to the anterior in this follicle, but I am unconvinced that the polar cells are mis-positioned based on the criteria given. Also, the authors state that this follicle has a single small polar cell cluster that contained both WT and *eya* mutant cells. Are the authors saying that a clone of *eya* cells in one part of the epithelium eliminates the normal polar cells on the other side? I understand why the authors wanted to do the *eya* experiment, but given the confusing results, it might be better to remove these data from the paper.

The paper makes extensive use of RNAi transgenes without any controls or references showing the specificity and/or effectiveness of these reagents. At the very least, the authors should confirm that the jak/stat pathway RNAis in Figure 2 reduce the expression of the 10xStatGFP reporter, and the extent to which their sqh RNAi reduces myosin levels via pSqh staining.

The authors mention a Gal4 driver that only drives expression at the follicle poles. In the text it is called Ft-Gal4, whereas in the figures it is called Fru-Gal4. Which notation is correct? Also the authors should either cite a reference showing that this driver is exclusive to the terminal domains or show it themselves with a UAS-GFP reporter.

The authors claim that Figure 2 shows a specific effect of a hop RNAi clone on the posterior half of the follicle. While the anterior is less round than the posterior by eAR analysis, it is far from normal, as the anterior typically has a sharp point at this stage, as shown in Figure 2. The authors should tone down their claims here.

In Figure 2, the authors show that over-expressing Upd in the center of the epithelium disrupts follicle elongation. If over-expressing Upd using the Upd-Gal4 driver is sufficient to hyper-elongate the follicle, this result would provide even stronger support for the authors' model. This is an easy experiment that should be attempted.

In Figure 5 it looks as if there are gaps in the epithelium in the *sqh* clone, a phenotype that was previously documented by Wang and Riechmann (2007). If this is true, it makes it very difficult to interpret the results of this experiment. The authors should confirm that the epithelium is fully intact for all follicles assayed.

*Reviewer #2:*

In this study the authors analyze the change in shape (elongation) that egg chambers experienced between the early stages (3-7). They also describe changes at the cell and tissue level that happen in that period, and try to understand the causal link between these cellular and tissue changes with the early elongation. It is an original and quite comprehensive study of morphological and molecular changes from stages 3-8, but there are some problems in the study and interpretation that make it difficult for me to believe that the causal link is actually established.

1) Most of the conclusions about the effects on elongation are based on the fact that stage 5-7 mutant egg chambers show a rounder shape than stage 5-7 wildtype egg chambers. Because of this, it is crucial that stage 5-7 are properly identified, especially in mutants, as a mis-identification of an early egg chamber (e.g., stage 4) for an older one (stage 6-7) would have a huge impact in the interpretation of the phenotypes. How are the stages defined? This point is important not only for the mutant egg chambers, but also for wildtype ones. How are the authors staging so precisely stage 3, stage 4, stage 5, and so on? It is not an easy task, as addressed in this paper Sci Rep. 2016 Jan 6;6:18850. doi: 10.1038/srep18850. Automatic stage identification of *Drosophila* egg chamber based on DAPI images. Jia D et al.)

In several experiments, a mutant egg chamber is defined to be one precise stage, but I doubt that these stagings could have been achieved morphologically, as the mutant egg chambers would also present defects in cell numbers, cell shape, cell fate, egg chamber shape, etc..– that would strongly affect the staging. The authors need to clarify how the staging was done, and also need to characterize stages by molecular markers. I would suggest starting with markers of terminal fate, as well as PH3 stainings, as this would show the mitotic state of cells (follicle cells exit mitosis at stage 6); Staufen, which starts being localized to the posterior at stage 7-8, and maybe for other ideas see the paper I referred to above.

For example: Figure 2 – this egg chamber is to me obviously younger that the control in A), as it can be seen by the shape of the oocyte anterior membrane (a V shape in C), while a straighter membrane in A)), but they are both defined as stage 7.

2) There is a clear correlation between polar cells (PCs) positioning and egg chamber early elongation, and this is an original thought, but I do not think the causal relationship has been proven, and I understand this to be a difficult task. Two of the best experiments for the possible causal link are in Figure 1 and Figure 2, and for this reason, it should be included a comprehensive description and quantification of the results: What percentage of clones show this phenotype and how many have been analyzed (this is specially missing in Figure 2)?

3) Regarding the myo2 function in elongation: The authors do a nice job in characterizing actomyo behavior in the apical membrane, and relating this to apical surface changes. I especially liked the original approach to filming the poles. However, the link with elongation is again a hard one to establish: since reducing myo2 activity results in such a huge effect on cell numbers, it is very difficult to conclude that it is the lack of myo2 what is responsible for the defects in elongation. I think the authors need to manipulate the myo2 pathway by other means that might have less of an impact in cell number. For example, manipulating the activity of Rho, Rock, myo2 phosphatases, Myo2LC, etc., I would also like to suggest that the activities of these components of the pathway are both reduced (ag., mutants, dominant negatives), as well as increased (over-expression, dominant active forms, etc.), when possible.

4) Regarding Jak/stat: since the interpretation is that the jak/stat gradient impacts on myo2 and then on elongation, and since the gradient is present in both anterior and posterior poles, the same findings described for posterior pole should be analyzed in anterior poles. It would be ideal to get an idea of the volume and apical surface changes in the anterior pole, as well as on the myo2 behavior there. Also, when studying the effect of the Jak/Stat pathway on the stage 3-7 cellular changes, they need to include analysis not only at stage 7, but stage 3-7. In fact, the described defects at stage 7 do not show much of an effect, and further analysis is required. For example, reducing upd in polar cells or inactivating jak/stat in terminal follicle cells at stage 7 should eliminate the high surface variation in both poles and stage 7 poles should then be similar to stage 7ML. The authors need to check this.

Also, does the jak/stat pathway impact on the gradient of actomyosin contraction at the poles? If possible this needs to be checked, as the surface variation gradient, which may be affected by jak/stat gradient, is not necessarily a complete reflection of actomyo contraction.

Furthermore, no experiment shows that the production of upd by the PCs is required, as there is no experiment showing than when this secretion is affected, elongation is aberrant. The authors need to eliminate, or at least reduce, upd in the PCs to answer this question. In the TjG4, updRNAi experiment, upd is reduced in all follicle cells, but I am not sure it is reduced on PCs, could this be described, please? And if the PCs are the source of upd, why is the reduction of upd in all other follicle cells giving a phenotype? And related to this, why would overexpression of upd in the follicle cells that are not the source, result in defects in elongation?

---

## [Author Response]

[Editors’ note: the author responses to the first round of peer review follow.]

[…] In the Abstract the authors write that elongation occurs "without planar cell polarity requirement". However, they showed lack of planar polarization just for MyoII and Baz (subsection “MyosinII activity drives apical pulses and early elongation”).

There are two ideas associated with this sentence. First, the elongation that we observed is independent of Fat2, which controls the only pcp pathway known the be active in this tissue. Second, apical MyoII is not planar polarized thought it is the motor of elongation. We prefer to maintain this sentence as it is in the Abstract because of the lack of space but we tried to give better explanations in the Discussion: “Moreover, the absence of planar polarization of MyoII in apical, the driving force of early elongation, and the nonrequirement for “basal pcp” strongly argues against a control of this elongation phase via a planar a cell polarity working at a local scale.”

Subsection “Polar cells define the axis of early elongation”, second paragraph: In Pak mutants, the authors often observe a single polar cell cluster (it should be Figure 1, not 1D) and conclude, that Pak is required for polar cell "positioning". If it is just involved in positioning, I would expect two follicle cell clusters in each case. For me it looks that Pak is required either for polar cell specification or for polar cell survival.

We agree with this comment. We initially wrote that it “suggests that *Pak* is required for polar cell positioning” because our observation of this phenotype suggests that there are initially two clusters per follicle but that the persistence of stalk cells along follicles (and not only between) that contact the clusters induces somehow their convergence and fusion. However, it is indeed overstated at the level of description that we provide and we therefore modified the sentence accordingly.

I also had some difficulties to find the link between the JAK-STAT activity gradient and the morphogenetic behavior (intercalation, stretching) described in the second part.

JAK-STAT induces a gradient of pulsing from the poles. These pulses induce an apical constriction at the poles, as shown by the reverse phenotype of STAT mutant (cell stretching) and in agreement with what happens in many tissues where pulses induce constriction. Then, the full understanding of the link between JAK-STAT and pulses will require to identify the transcriptional targets of STAT in this tissue and analyse their function. We will definitively try this in the future.

The link between JAK-STAT and intercalations is indeed more difficult to establish. We mentioned these intercalations because they probably explain why the cells are not stretched in the AP axis whereas the follicle elongates. However, there is no indication that they are instrumental for elongation, and they might rather reflect a passive response of the tissue to relax the tensions due to the elongation. Thus, we assume that demonstrating a link between JAK-STAT and intercalation is not essential for our work at this stage.

Reviewer #1:[…] Statistical analyses need to be reported for the graph in Figure 1. This will likely require increasing the n for many of the fat2 measurements as some of them are currently very low (including one stage that is zero).

We have increased our sample size and performed a proper statistical analysis. The stage with 0 follicle in *fat2* mutant was due to the fact that the WT stages 10A and 10B were plotted separately. We apologise for this mistake.

Figure 1 report aspect ratios using a range of stages (4-8). The range is problematic. If one group has more stage 8 follicles and the other has more stage 4 follicles, it is possible to obtain a false positive result. These types of comparisons can only be made when all of the follicles are at a single stage.

This representation was initially chosen because, in both cases, one of the two categories were very rarely observed and it was therefore not possible to provide statistics for each stage. Nonetheless, we agree with this formal criticism, though we had it in mind and paid attention that it actually did not apply to our samples.

We therefore significantly increased our sample size. However, it is still impossible to plot these follicles according to their stages and performed a statistical test for each stage and each category. Rather, we plotted the long axis as a function of the short axis, a method that offers the advantage of no bias due to stage determination (see response to point #1 reviewer #2), and statistically compared the slope of the regression lines, which we defined as “elongation coefficient” (Figure 1 and Figure 1—figure supplement 1). This approach confirms our previous conclusions.

Figure 1 purports to show that the polar cells are mis-positioned because they are not adjacent to the stalk. However, the stalk moves away from the polar cells during mid-oogenesis, a phenomenon that can be seen in the oldest follicle in Figure 1. The oocyte appears to be mis-localized to the anterior in this follicle, but I am unconvinced that the polar cells are mis-positioned based on the criteria given. Also, the authors state that this follicle has a single small polar cell cluster that contained both WT and eya mutant cells. Are the authors saying that a clone of eya cells in one part of the epithelium eliminates the normal polar cells on the other side? I understand why the authors wanted to do the eya experiment, but given the confusing results, it might be better to remove these data from the paper.

We agree that our data on *eya* were much more complicated than expected, based on published description of the phenotype. We therefore also agree to remove these data. Also, we added the effect on elongation of Upd:gal4, Upd:RNAi, which, has a significant effect on early elongation (Figure 2). Upd:gal4 being only expressed in polar cells (Khammari et al., 2011), this result causally links polar cells and early elongation.

The paper makes extensive use of RNAi transgenes without any controls or references showing the specificity and/or effectiveness of these reagents. At the very least, the authors should confirm that the jak/stat pathway RNAis in Figure 2 reduce the expression of the 10xStatGFP reporter, and the extent to which their sqh RNAi reduces myosin levels via pSqh staining.

We performed these controls with STAT-GFP for STAT92E and Upd RNAi and observed a strong diminution of the signal (Figure 2—figure supplement 1). Surprisingly, we did not observed an effect of the RNAi against Hop on STAT-GFP. We checked the identity of this line by sequencing and it was ok. It is therefore unclear why this line induced the same kind of elongation and pulsing defects than STAT or Upd RNAi but has no effect on STAT-GFP. We therefore removed all the data obtained with Hop RNAi and replaced all with experiments with STAT92E and/or Upd RNAi, with the exception of the previous Figure 2. Importantly it did not change our conclusions. We also looked at pSqh in sqh RNAi and it is strongly reduced (Figure 3—figure supplement 1).

The authors mention a Gal4 driver that only drives expression at the follicle poles. In the text it is called Ft-Gal4, whereas in the figures it is called Fru-Gal4. Which notation is correct? Also the authors should either cite a reference showing that this driver is exclusive to the terminal domains or show it themselves with a UAS-GFP reporter.

Fru is the proper abbreviation, sorry for the mistake. Fru:Gal4 expression profile is described in Borensztejn et al., 2013, and we now mentioned it in the text.

The authors claim that Figure 2 shows a specific effect of a hop RNAi clone on the posterior half of the follicle. While the anterior is less round than the posterior by eAR analysis, it is far from normal, as the anterior typically has a sharp point at this stage, as shown in Figure 2. The authors should tone down their claims here.

This figure corresponded to a flip-out experiment, in which the Hop knocked-down cells are marked by the GFP and are therefore at the anterior, and not the posterior of the follicle. However, this experiment, obtained with Hop RNAi, has been removed. We tried to reproduce it with stat RNAi but we did not manage to find proper conditions.

Nonetheless, it is worth noticing that we have other results indicating that JAK-STAT acts in cell autonomous manner at each pole. STAT has a cell-autonomous effect on apical surface, cell height and phospho-MyoII recruitment (Figure 5 and Figure 6) and MyoII is specifically required at each pole for elongation (Figure 5).

In Figure 2, the authors show that over-expressing Upd in the center of the epithelium disrupts follicle elongation. If over-expressing Upd using the Upd-Gal4 driver is sufficient to hyper-elongate the follicle, this result would provide even stronger support for the authors' model. This is an easy experiment that should be attempted.

We tried to overexpress Upd with Upd Gal4 but did not observed any effect on elongation. Of notice, we placed STAT-GFP reporter in such flies and did not observed any change in JAK-STAT activation gradient. Thus, this result suggests a mechanism buffering Upd production by polar cells, which might be interesting to look at in the future but is beyond the scope of this article on tissue elongation.

In Figure 5 it looks as if there are gaps in the epithelium in the sqh clone, a phenotype that was previously documented by Wang and Riechmann (2007). If this is true, it makes it very difficult to interpret the results of this experiment. The authors should confirm that the epithelium is fully intact for all follicles assayed.

We agree that gaps in the epithelium could be an issue. It was actually why, for this experiment, we stained for Dlg and not just F-actin. It allowed us to get a better view of the small lateral domains of the cells, and follicles with gaps were excluded from the analysis. This point is now specified in the methods.

Of notice, we now provide data obtained with *rok* mutant clones, in which the cells are less flatten and gaps are not observed, with the same effect on elongation than with *sqh* clones.

Reviewer #2:In this study the authors analyze the change in shape (elongation) that egg chambers experienced between the early stages (3-7). They also describe changes at the cell and tissue level that happen in that period, and try to understand the causal link between these cellular and tissue changes with the early elongation. It is an original and quite comprehensive study of morphological and molecular changes from stages 3-8, but there are some problems in the study and interpretation that make it difficult for me to believe that the causal link is actually established.1) Most of the conclusions about the effects on elongation are based on the fact that stage 5-7 mutant egg chambers show a rounder shape than stage 5-7 wildtype egg chambers. Because of this, it is crucial that stage 5-7 are properly identified, especially in mutants, as a mis-identification of an early egg chamber (e.g., stage 4) for an older one (stage 6-7) would have a huge impact in the interpretation of the phenotypes. How are the stages defined? This point is important not only for the mutant egg chambers, but also for wildtype ones. How are the authors staging so precisely stage 3, stage 4, stage 5, and so on? It is not an easy task, as addressed in this paper Sci Rep. 2016 Jan 6;6:18850. doi: 10.1038/srep18850. Automatic stage identification of Drosophila egg chamber based on DAPI images. Jia D et al.)

We have indeed seen this very interesting method and we will try to use it in the future. The first figure of this article actually nicely sums up what was already described in Spradling, 1993 (based on King 1970) in which stage definitions is only based on white light (allowing seeing the presence of vitellus or not) and DAPI staining. We used the same criteria as indicated in the Materials and methods.

In several experiments, a mutant egg chamber is defined to be one precise stage, but I doubt that these stagings could have been achieved morphologically, as the mutant egg chambers would also present defects in cell numbers, cell shape, cell fate, egg chamber shape, etc. – that would strongly affect the staging.

As mentioned previously, stage definition was assessed by criteria that are independent of cell number, cell shape or egg chamber shape.

The authors need to clarify how the staging was done, and also need to characterize stages by molecular markers. I would suggest starting with markers of terminal fate, as well as PH3 stainings, as this would show the mitotic state of cells (follicle cells exit mitosis at stage 6); Staufen, which starts being localized to the posterior at stage 7-8, and maybe for other ideas see the paper I referred to above.

DAPI staining is sufficient to detect the arrest of cell division. Moreover, it would be impossible to define a combinatory of markers that allows the unambiguous determination of all the stages. Finally, as mentioned before, the original definition of the stages has been done without any markers excepted DAPI.

However, we understand your concern and we therefore conducted two simple tests:

1) We compared all the mutant genotypes at a stage n to stage n-1 wild-type follicles. Doing so, there is still a significant difference for most of tested genotypes, including actors of the JAK-STAT pathway and MyosinII. In other words, even though we had systematically underestimated the stage of mutant follicles and not the wild-type ones, which I am sure it is not the case, the main conclusions would not be affected.

2) More significantly, we plot the long axis as a function of the short axis for previtellogenic stages and make the regression line as shown on Figure 1—figure supplement 1. We determined the slope of the lines which corresponds to what we defined as an “elongation coefficient”. Importantly, this method is independent of stage determination. We validated this method using *fat2* mutant (Figure 1—figure supplement 1). Statistical comparison of the elongation coefficients confirmed our previous results (Figure 1, Figure 2, Figure 3).

For example: Figure 2 – this egg chamber is to me obviously younger that the control in A), as it can be seen by the shape of the oocyte anterior membrane (a V shape in C), while a straighter membrane in A)), but they are both defined as stage 7.

We would have the same visual impression based on V shape versus straight membrane. However, both can be observed at stage 7 (i.e. WT controls on Figure 1 and Figure 2), even on the same follicle at 2µm of differences in the z axis as depicted on pictures of a same follicle (Author response image 1).

2) There is a clear correlation between polar cells (PCs) positioning and egg chamber early elongation, and this is an original thought, but I do not think the causal relationship has been proven, and I understand this to be a difficult task. Two of the best experiments for the possible causal link are in Figure 1 and Figure 2, and for this reason, it should be included a comprehensive description and quantification of the results: What percentage of clones show this phenotype and how many have been analyzed (this is specially missing in Figure 2)?

As proposed by reviewer #1, to avoid any confusion due to these results, *eya* data were removed. Figure 2 was performed with Hop RNAi and was also therefore removed. We tried to reproduce such phenotype with STAT RNAi but for some reasons probably linked to the role of JAK-STAT during follicle budding, we could not defined proper conditions. However, thanks to a later suggestion, we now provided data that clearly link polar cells and elongation. Upd is only expressed in polar cells and Upd:Gal4 reproduces this pattern (Khammari et al., 2011). Knocking-down Upd with this driver affects early elongation and thus polar cells are required for early elongation.

3) Regarding the myo2 function in elongation: The authors do a nice job in characterizing actomyo behavior in the apical membrane, and relating this to apical surface changes. I especially liked the original approach to filming the poles. However, the link with elongation is again a hard one to establish: since reducing myo2 activity results in such a huge effect on cell numbers, it is very difficult to conclude that it is the lack of myo2 what is responsible for the defects in elongation. I think the authors need to manipulate the myo2 pathway by other means that might have less of an impact in cell number. For example, manipulating the activity of Rho, Rock, myo2 phosphatases, Myo2LC, etc., I would also like to suggest that the activities of these components of the pathway are both reduced (ag., mutants, dominant negatives), as well as increased (over-expression, dominant active forms, etc.), when possible.

The question of the primary effect of MyoII on elongation is indeed really important. First of all, one could emphasize that we already affected MyoII by two different means:

- A drastic one with a null mutant, but in clones, which allows us to show a specific role of MyoII at the poles (Figure 5), which is coherent with the proposed role of the pulses in these regions.

- A more subtitle one, with a RNAi in the whole epithelium, which only moderately affects cell shape but still shows an elongation defect (Figure 4).

Nonetheless, as suggested, we also looked at the involvement of the Rho kinase using null mutant clones. These mutant clones recapitulate the effect on elongation observed with *sqh*, though the flattening of the mutant cells was weaker (Figure 5). Thus, this result confirms both the involvement of the Rho pathway and the primary effect of this pathway on early elongation.

The MyoII gain of function was a very interesting suggestion. We tried by overexpression of the phosphomimetic form Sqh^EE^ (and Sqh^DD^) in all the follicle cells, with tj:gal4, or at the poles, with Fru:gal4. In both cases we observed no effect on tissue elongation. These results are actually consistent with the fact the Sqh phosphorylation is homogenous along AP axis whereas pulses are not. Thus, the spatiotemporal control of the pulses by JAKSTAT is not at the level of Myosin II activation and it is therefore logical that Sqh phosphorylation is necessary but not sufficient to modulate tissue elongation. Alternatively, the absence of effect could be due to the fact that these mutant forms of Sqh are not active as proper phosphorylated protein as it has been very recently shown (Vasquez et al., *eLife*, 2016). Thus, such experiments appear inconclusive and we did not add them in the new version of the article, but it could be done upon request.

An alternative approach for a gain of function of MyoII would have been to use a mutant for the myosin phosphatase. However, it has been recently shown that such mutant follicle cells prematurely undergo basal polarized oscillations (Valencia-Expósito, Nat Com, 2016), which normally start at stage 9 and are likely the driving force for the second elongation phase (He et al., Nat Cell Biol 2010). Consistently, we observed that this mutation leads to a dramatic increase of pSqh staining at the basal domain but not at the apical one during early stages (see the two examples in Author response image 2). The effect on elongation of such mutant might be therefore confusing and was not analysed.

**Author response image 2. respfig2:** 

4) Regarding Jak/stat: since the interpretation is that the jak/stat gradient impacts on myo2 and then on elongation, and since the gradient is present in both anterior and posterior poles, the same findings described for posterior pole should be analyzed in anterior poles. It would be ideal to get an idea of the volume and apical surface changes in the anterior pole, as well as on the myo2 behavior there.

We agree that it would be ideal but it turn out that the trick that we used to film the poles cannot be applied to the anterior pole for technical reasons. Each ovariole is initially in a muscle sheath that has to be removed for imaging. To do so, we pull on the anterior tip of an ovariole whereas we maintain the rest of the ovary. Either it leads to the complete removal of the ovariole from the muscle sheath, or it leads to a break of an interfollicular filament and we release only part of an ovariole, with the posterior pole of the last follicle that can be easily observed when it is placed between beads. However, such approach does not allow to obtain “free” anterior poles. The solution would be a mechanical dissection between follicles but it could be then almost impossible to confidently assess whether the poles would be damaged or not.

Therefore, the only new data that we can provide about the anterior pole is the height of the cells. It is also increased, compared to younger stages and to lateral cells, as observed at the posterior (Figure 6). Since an increase in cell height is classically associated with cell constriction and since both JAK-STAT and MyoII affect cell height in the follicular cells, we assume that similar events are taking place at each poles.

Also, when studying the effect of the Jak/Stat pathway on the stage 3-7 cellular changes, they need to include analysis not only at stage 7, but stage 3-7. In fact, the described defects at stage 7 do not show much of an effect, and further analysis is required. For example, reducing upd in polar cells or inactivating jak/stat in terminal follicle cells at stage 7 should eliminate the high surface variation in both poles and stage 7 poles should then be similar to stage 7ML. The authors need to check this.Also, does the jak/stat pathway impact on the gradient of actomyosin contraction at the poles? If possible this needs to be checked, as the surface variation gradient, which may be affected by jak/stat gradient, is not necessarily a complete reflection of actomyo contraction.

We agree that it is an important point. We now provide data showing that Upd RNAi:

- reduces pulse intensity at the poles at stage 7-8 (Figure 4)

- increases cell surface at the poles at stage 7-8 (Figure 6)

Furthermore, no experiment shows that the production of upd by the PCs is required, as there is no experiment showing than when this secretion is affected, elongation is aberrant. The authors need to eliminate, or at least reduce, upd in the PCs to answer this question. In the TjG4, updRNAi experiment, upd is reduced in all follicle cells, but I am not sure it is reduced on PCs, could this be described, please? And if the PCs are the source of upd, why is the reduction of upd in all other follicle cells giving a phenotype? And related to this, why would overexpression of upd in the follicle cells that are not the source, result in defects in elongation?

As mentioned before, Upd, and Upd:Gal4 are expressed only in the polar cells whereas Tj:gal4 is expressed in all FCs including PCs. We now provided experiments with Upd:gal4 and we obtained similar phenotype.From our point of view, once we have shown that Upd is required for early elongation, the fact that its ectopic expression blocks elongation is actually the best argument that it needs to be expressed in a specific pattern and thus delivering a spatial cue for elongation.